# Designed peptides as nanomolar cross-amyloid inhibitors acting via supramolecular nanofiber co-assembly

Karin Taş [1], Beatrice Dalla Volta[1], Christina Lindner[1,10], Omar El Bounkari[2], Kathleen Hille[1], Yuan Tian[2], Xènia Puig-Bosch[3], Markus Ballmann[3], Simon Hornung [1], Martin Ortner [1,11], Sophia Prem[1,12], Laura Meier[4], Gerhard Rammes[3], Martin Haslbeck[4], Christian Weber[5,6,7,8], Remco T. A. Megens [5,7,9], Jürgen Bernhagen [2,6] & Aphrodite Kapurniotu [1] ✉

Amyloid self-assembly is linked to numerous devastating cell-degenerative diseases. However, designing inhibitors of this pathogenic process remains a major challenge. Cross-interactions between amyloid-β peptide (Aβ) and islet amyloid polypeptide (IAPP), key polypeptides of Alzheimer's disease (AD) and type 2 diabetes (T2D), have been suggested to link AD with T2D pathogenesis. Here, we show that constrained peptides designed to mimic the Aβ amyloid core (ACMs) are nanomolar cross-amyloid inhibitors of both IAPP and Aβ42 and effectively suppress reciprocal cross-seeding. Remarkably, ACMs act by co-assembling with IAPP or Aβ42 into amyloid fibril-resembling but non-toxic nanofibers and their highly ordered superstructures. Co-assembled nanofibers exhibit various potentially beneficial features including thermolability, proteolytic degradability, and effective cellular clearance which are reminiscent of labile/reversible functional amyloids. ACMs are thus promising leads for potent anti-amyloid drugs in both T2D and AD while the supramolecular nanofiber co-assemblies should inform the design of novel functional (hetero-) amyloid-based nanomaterials for biomedical/biotechnological applications.

Amyloid self-assembly is linked to numerous devastating cell-degenerative diseases, with AD and T2D being two of the most prominent ones[1,2]. The main component of amyloid plaques in AD brains is the 40(42)-residue peptide Aβ40(42), while pancreatic amyloid of T2D patients consists of fibrillar assemblies of the 37-residue IAPP[2,3] (Fig. 1a). IAPP is secreted from pancreatic β-cells and functions as a neuroendocrine regulator of glucose homeostasis[3]. However, the formation of cytotoxic IAPP

[1]Division of Peptide Biochemistry, TUM School of Life Sciences, Technical University of Munich (TUM), 85354 Freising, Germany. [2]Division of Vascular Biology, Institute for Stroke and Dementia Research (ISD), Klinikum der Universität München, Ludwig-Maximilian-University (LMU), 81377 Munich, Germany. [3]Department of Anesthesiology and Intensive Care, Technical University of Munich/Klinikum Rechts der Isar, 81675 München, Germany. [4]Center for Protein Assemblies, Department of Chemistry, Technical University of Munich, 85748 Garching, Germany. [5]Institute for Cardiovascular Prevention, Klinikum der Universität München, Ludwig-Maximilian-University Munich (LMU), 80336 Munich, Germany. [6]Munich Cluster for Systems Neurology (SyNergy), 81377 Munich, Germany. [7]German Centre for Cardiovascular Research (DZHK), partner site Munich Heart Alliance, 80802 Munich, Germany. [8]Department of Biochemistry, Cardiovascular Research Institute Maastricht (CARIM), Maastricht University, 6229 Maastricht, The Netherlands. [9]Department of Biomedical Engineering, Cardiovascular Research Institute Maastricht (CARIM), Maastricht University, 6229 Maastricht, The Netherlands. [10]Present address: Institute of Organic Chemistry, Centre for Advanced Materials, 69120 Heidelberg, Germany. [11]Present address: Chair of Biopolymer Chemistry, TUM School of Life Sciences, Technical University of Munich (TUM), 85354 Freising, Germany. [12]Present address: Werner Siemens-Chair of Synthetic Biotechnology (WSSB), Technical University of Munich (TUM), Department of Chemistry, 85748 Garching, Germany. ✉e-mail: akapurniotu@mytum.de

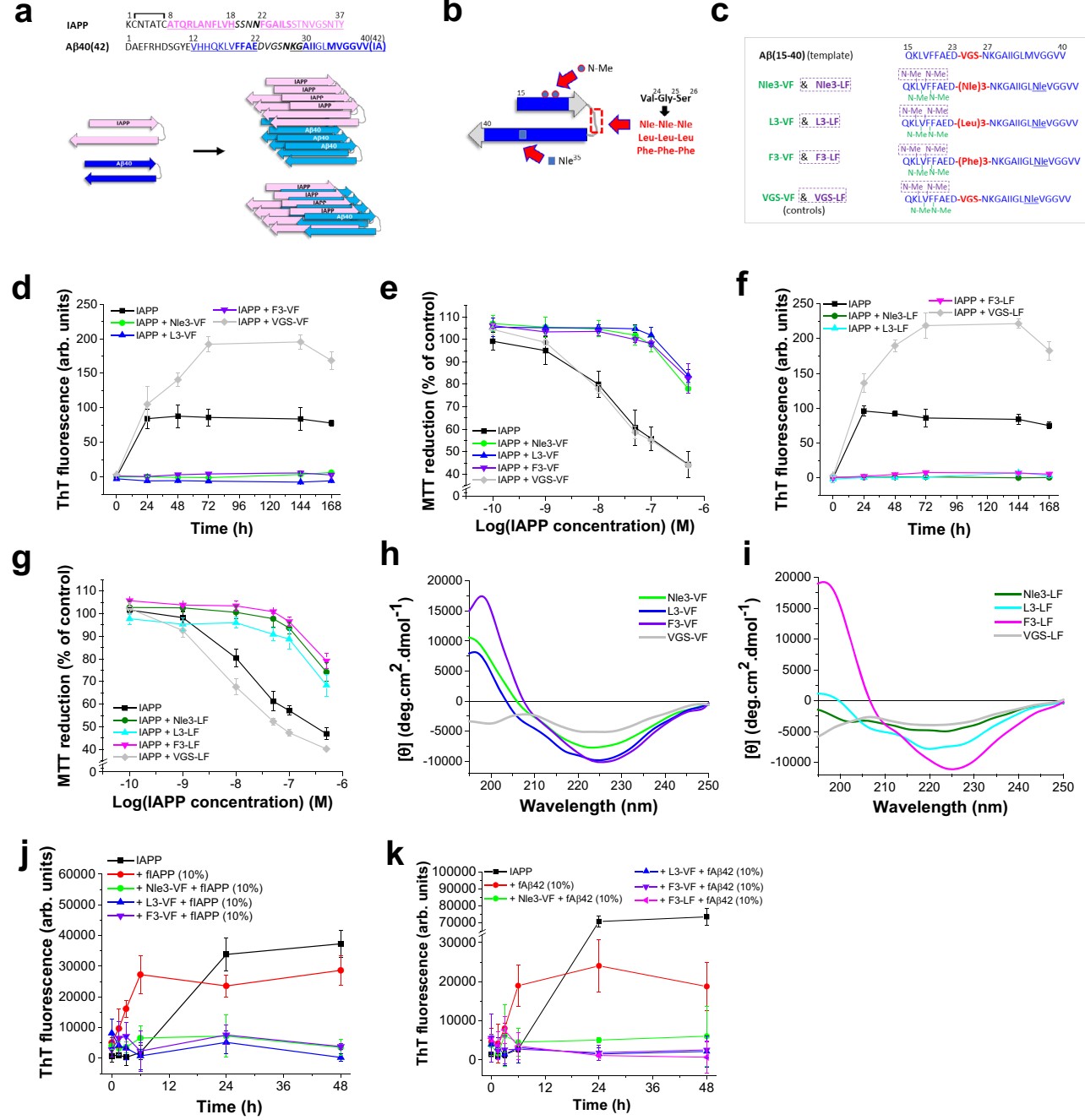

**Fig. 1 | ACM design concept, their effects on IAPP amyloid self-assembly and cytotoxicity, and ACM secondary structures. a** Sequences of IAPP and Aβ40(42), proposed models of fIAPP and fAβ40 folds, and hypothetical IAPP/Aβ40 "hetero-amyloids" (β-strands, pink or blue and underlined; "hot segments" of self-/cross-interactions, bold; loop residues, italics)[25,30,31]. **b** ACM inhibitor design strategy. Template Aβ(15–40) in a β-strand-loop-β-strand fold proposed for fAβ40[31] is modified via (**a**) N-methylations in Aβ(17–20), **b** substitution of Aβ(24-26) by hydrophobic tripeptides, and **c** Met35 substitution by Nle. **c** Sequences of the six ACMs and negative controls VGS-VF and VGS-LF (Supplementary Table 1). Each sequence corresponds to two different ACMs which contain the same LTS but a different couple of N-methylated residues (dashed boxes). Color code as in **a**; LTS and tripeptide VGS in red; green or violet for peptide names and corresponding N-methylated residues. **d** Nle3-VF, L3-VF, and F3-VF block IAPP amyloid self-assembly. Fibrillogenesis of IAPP (16.5 µM) alone or with ACMs or VGS-VF was assessed via ThT binding (IAPP/peptide 1/2) (means ± SD, $n = 3$ independent assays). **e** Nle3-VF, L3-VF, and F3-VF suppress the formation of toxic IAPP

assemblies. Solutions of **d** (7-day-aged (VFS-VF 24 h)) added to RIN5fm cells; cell viability determined via MTT reduction (means ± SD, three independent assays, $n = 3$ technical replicates each). **f** Nle3-LF, L3-LF, and F3-LF block IAPP amyloid self-assembly. Assay as in **d** (IAPP/peptide 1/2 except L3-LF (1/2.5)) (means ± SD, three independent assays). **g** Nle3-LF, L3-LF, and F3-LF suppress the formation of toxic IAPP assemblies. Solutions of **f** (7-day-aged (VGS-LF 24 h)) added to RIN5fm cells; cell viability determined via MTT reduction (means ± SD, three independent assays, $n = 3$ technical replicates each). **h, i** Secondary structure of ACMs. Far-UV CD spectra of ACMs of **d** and **f** versus non-inhibitors (5 µM, pH 7.4). **j** ACMs inhibit seeding of IAPP by preformed fIAPP. Fibrillogenesis of IAPP (12 µM) without or with fIAPP seeds (10%) and seeded IAPP/ACM mixtures assessed via ThT binding (IAPP/ACM 1/2) (means ± SD, $n = 9$ (for IAPP alone) or 3 (for all other samples) independent assays). **k** ACMs inhibit fAβ42-mediated cross-seeding of IAPP. Fibrillogenesis of IAPP with and without fAβ42 seeds (10%) versus IAPP/ACM mixtures (IAPP 12 µM, IAPP/ACM 1/2) (means ± SD, $n = 6$ (for IAPP with or without seeds) or $n = 3$ (for all other samples) independent assays).

assemblies and amyloid fibrils mediates pancreatic β-cell degeneration in T2D[3].

Epidemiological studies suggest that T2D patients have an increased risk of AD and vice versa[4–7]. In addition, increasing evidence suggests molecular and pathophysiological links between both diseases[7–10]. Cross-interactions between Aβ and IAPP could be such molecular links[7–16]. In fact, polymorphic Aβ/IAPP interactions are able to cross-seed or cross-suppress amyloid self-assembly depending on structures and self-assembly states of the interacting polypeptides[7–9,11–13,16]. To this end, IAPP and Aβ fibrils act as reciprocal cross-seeds of amyloid self-assembly, as shown by both in vitro and experimental in vivo studies[8,9,11]. On the other hand, nanomolar affinity interactions between early prefibrillar and non-toxic IAPP and Aβ species redirect both polypeptides into initially non-fibrillar and non-toxic co-assemblies, thus delaying amyloid self-assembly[12,13]. Importantly, Aβ and IAPP were found to colocalize in AD- and T2D-related amyloid deposits both in humans and in mouse models[8–10,14,15]. Aβ/IAPP cross-interactions and putative "hetero-amyloids" could thus be highly relevant to the pathogenesis of both diseases[7–12,15,17,18].

Based on the above, molecules targeting amyloid self-assembly and reciprocal cross-seeding effects of IAPP and Aβ could be promising leads for anti-amyloid treatments in both AD and T2D[7,19]. However, so far, only a few inhibitors of amyloid self-assembly of both polypeptides (termed "cross-amyloid" inhibitors) have been reported and none of them suppressed reciprocal Aβ/IAPP cross-seeding[12,19–24]. Moreover, except for a recently approved and controversially discussed anti-Aβ amyloid antibody, no anti-amyloid treatments for AD or T2D have yet reached the clinic.

One reason for the high-affinity IAPP/Aβ40(42) cross-interactions could be the sequence similarity (50%) and identity (~25%) between both polypeptides (Fig. 1a)[11,25]. Notably, highest degrees of sequence identity/similarity are observed between their amyloid core segments IAPP(8–28) and Aβ(15–40(42)). In addition, the same IAPP- or Aβ40(42)-"hot segments" within their amyloid core segments were found to mediate both self- and cross-interactions (Fig. 1a)[11,25,26]. Strong similarities exist also between their fibril folds and potential cross-seeding interfaces within putative hetero-amyloids were proposed[13,24,25,27–31].

Capitalizing on IAPP/Aβ cross-interactions, we have previously designed peptides derived from the IAPP amyloid core IAPP(8–28) as IAPP "interaction surface mimics" (ISMs)[20]. ISMs effectively suppressed amyloid self-assembly of Aβ40(42) and/or IAPP by sequestering them into amorphous, non-toxic aggregates[20].

Here, we explored the idea of designing peptides derived from the Aβ40 amyloid core Aβ(15–40) as Aβ "amyloid core mimics" (ACMs) and inhibitors of amyloid self-assembly and cross-seeding interactions of IAPP and Aβ42. Our inhibitor design concept aimed at distorting the pathogenic fibril fold of Aβ(15–40) and stabilize alternative, amyloid-like but non-amyloidogenic folds[19]. These should yield alternative interaction surfaces with IAPP or Aβ42 and redirect them into non-fibrillar and non-toxic aggregates[12,19,20,32,33]. A series of conformationally constrained peptides was synthesized and studied. In fact, ACMs were non-amyloidogenic and non-cytotoxic, bound IAPP and Aβ42 with nanomolar affinity, and fully blocked their cytotoxic amyloid self-assembly. Furthermore, ACMs effectively suppressed reciprocal cross-seeding effects. Surprisingly, ACMs exerted their inhibitory function by co-assembling with IAPP or Aβ42 into amyloid fibril-resembling nanofibers and their diverse, highly ordered superstructures. For their characterization, a spectrum of biophysical, biochemical, and advanced microscopy methods, including confocal laser-scanning microscopy (CLSM), stimulated emission depletion (STED) imaging, two-photon microscopy (2PM), and fluorescence lifetime imaging microscopy (FLIM)-based Förster resonance energy transfer (FRET) (FLIM-FRET) was applied. In addition, in vitro and ex vivo cell-based assays were used. In strong contrast to IAPP or Aβ42 fibrils (fIAPP or fAβ42), co-assembled nanofibers were "ThT-invisible", non-cytotoxic, and seeding-incompetent. Moreover, they were thermolabile, easily degradable by proteinase K (PK), and became efficiently phagocytosed in vitro by primary macrophages and cultured microglial cells.

## Results

### Inhibitor design and concept evaluation

For inhibitor design, Aβ(15–40) was used as a template in the context of the fAβ40 fold suggested by Petkova et al.[31,34], which features a β-strand-loop-β-strand motif with Aβ(12–22) and Aβ(30–40) forming the β-strands and Aβ(23–29) the loop (Fig. 1a, b). Of note, this U-shaped fold has often been applied to model Aβ-IAPP hetero-amyloids[35,36]. A minimum number of chemical modifications was made aiming at (a) distorting the loop, (b) stabilizing β-sheet structure, and (c) suppressing intrinsic amyloidogenicity of Aβ(15–40) while maintaining its pronounced self-/cross-assembly propensity in analogy to the ISM concept (Fig. 1b)[12,20,25,32]. The modifications were: (a) substitution of loop tripeptide Aβ(24–26) (Val-Gly-Ser) by β-sheet-propagating tripeptides consisting of identical large hydrophobic residues, which were expected to strengthen β-sheet interaction surfaces while being incompatible with localization in turns/β-arcs[37–39] and (b) selective amide bond N-methylation of two alternate residues within one of the two Aβ β-strand segments, which should suppress intrinsic amyloidogenicity of ACMs and their co-assemblies (Fig. 1b)[32,40,41]. Positions of N-methylations were based on fAβ40 models and previous SAR studies[31,34,40–42]. Finally, Met35 was replaced by Nle to avoid Met(O)-related side effects.

To evaluate the concept, 13 Aβ(15–40) analogs containing various different "loop tripeptide segments" (LTS), comprising (Nle)3, (Leu)3, (Phe)3, (Arg)3, (Gly)3, or Val-Gly-Ser (control LTS) and one pair of two N-methylated residues were designed, synthesized and studied (Fig. 1c and Supplementary Table 1). In addition, to identify best-suited LTS, various non-N-methylated analogs were synthesized and screened in initial studies (Supplementary Table 1): First, the effect of unmodified Aβ(15–40) (abbreviated VGS) on IAPP fibrillogenesis was studied by using the amyloid-specific thioflavin T binding assay and was found unable to inhibit (Supplementary Fig. 1a, b). However, non-N-methylated analogs Nle3 and L3 containing LTS (Nle)3 or (Leu)3 instead of Aβ(24–26) led to some delay in fibrillogenesis (Supplementary Fig. 1a, b). By contrast, analogs R3 and G3 containing LTS (Arg)3 and (Gly)3, respectively, did not inhibit and far-UV CD spectroscopy indicated less β-sheet structure than in Nle3 and L3 (Supplementary Fig. 1a–c). These findings suggested that Nle3 or L3 might be suitable candidates for further modifications.

Peptide Nle3 was then used as a template to identify best-suited positions for N-methylations. The four Nle3 analogs Nle3-LF, Nle3-VF, Nle3-GI, and Nle3-GG were synthesized, each of them containing two N-methylations placed at specific residues either within the N-terminal region corresponding to Aβ(15–23) (analogs Nle3-VF and Nle3-LF) or within the C-terminal region corresponding to Aβ(27–40) (analogs Nle3-GI & Nle3-GG) (Supplementary Fig. 2a and Supplementary Table 1). Peptides Nle3-VF and Nle3-LF (Fig. 1c) carrying N-methylations at Val18/Phe20 and at Leu17/Phe19, respectively, fully suppressed IAPP fibrillogenesis and cytotoxicity as determined by ThT binding in combination with the 3-[4,5 dimethylthiazol-2-yl]-2,5-diphenyltetrazolium bromide (MTT) reduction assay in cultured rat insulinoma (RIN5fm) cells (Fig. 1d–g, Supplementary Fig. 2b, c, and Supplementary Fig. 3a, b). By contrast, Nle3-GI and Nle3-GG, carrying N-methylations at Gly29/Ile31 and at Gly29/Ile33, respectively, did not inhibit (Supplementary Fig. 2b, c). Titrations of cytotoxic IAPP with Nle3-VF or Nle3-LF revealed nanomolar $IC_{50}$ values consistent with highly potent inhibitory activities (Table 1 and Supplementary Fig. 4a, b). Of note, the introduction of the N-methylations of Nle3-VF and Nle3-LF into the non-inhibitory peptides VGS, G3 or R3 did not convert them into

**Table 1 | IC$_{50}$ values of inhibitory effects of ACMs on IAPP-mediated cell damage[a] and app. $K_D$s of ACM/IAPP interactions[b][c]**

| ACM | IC$_{50}$ (±SD) (nM)[a] | app. $K_D$ (±SD) (nM)[b][c] |
|---|---|---|
| Nle3-VF | 65.0 (±5.2) | 69.5 (±1.4) |
| Nle3-LF | 82.1 (±10.2) | 55.4 (±5.9) |
| L3-VF | 112.5 (±8.1) | 77.3 (±2.9) |
| L3-LF | 133.2 (±29.0) | 143.2 (±5.0) |
| F3-VF | 78.5 (±13.6) | 15.0 (±1.9) |
| F3-LF | 41.7 (±4.1) | 37.6 (±2.9) |

[a]IC$_{50}$ values, means (±SD) from three independent titration assays ($n = 3$ technical replicates each).
[b]Determined by titrations of N-terminal fluorescein-labeled IAPP (5 nM; pH 7.4) with ACMs.
[c]App. $K_D$s are means (±SD) from three binding curves derived from three independent assays.

inhibitors (Supplementary Fig. 2d, e). Also, partial Nle3-VF segments Nle3-VF(15–23), Nle3-VF(27–40), and Nle3-VF(21–40) did not inhibit (Supplementary Fig. 2f–h). These results showed that both the loop tripeptide (Nle)3 and one of the two N-methylation patterns within Aβ(17–20) Val18Phe20 and Leu17Phe19 are required to convert Aβ(15–40) into a nanomolar inhibitor of IAPP.

To further evaluate the concept, we next synthesized and tested the four peptides L3-VF, L3-LF, F3-VF, and F3-LF containing loop tripeptides (Leu)3 or (Phe)3 and each of the two identified N-methylation patterns (Fig. 1c and Supplementary Table 1). All of them fully suppressed IAPP fibrillogenesis and related cytotoxicity (Fig. 1d–g and Supplementary Fig. 3c–f) and nanomolar IC$_{50}$ values were obtained (Table 1 and Supplementary Fig. 4c–f).

Far-UV CD spectroscopy revealed significant amounts of β-sheet structure in all six inhibitory ACMs whereas non-inhibitors VGS-VF and VGS-LF were less structured (Fig. 1h, i). In addition, ACMs exhibited strong self-assembly propensities as expected; however, they were soluble, non-amyloidogenic, and non-cytotoxic up to at least 500-fold higher concentrations than their IC$_{50}$ values (Supplementary Fig. 5). Importantly, in the presence of ACMs both self- and cross-seeding of IAPP amyloid formation by preformed fIAPP or fAβ42 were strongly suppressed as found by the ThT binding assay (Fig. 1j, k).

Together, our studies identified the six ACMs Nle3-VF, Nle3-LF, L3-VF, L3-LF, F3-VF, and F3-LF (Fig. 1c and Supplementary Table 1) as highly potent inhibitors of self-/cross-seeded IAPP amyloid self-assembly.

## ACMs co-assemble with IAPP into amyloid fibril-resembling but non-toxic nanofibers and their diverse highly ordered superstructures

To obtain insight into the inhibition mechanism, we next studied ACM interactions and co-assemblies. First, fluorescence spectroscopic titrations of N-terminal fluorescein-labeled IAPP (Fluos-IAPP) (5 nM) with ACMs revealed high-affinity interactions. In fact, most app. $K_D$s were <100 nM and in very good agreement with the determined IC$_{50}$ values (Fig. 2a, Supplementary Fig. 6, and Table 1). As freshly made solutions of Fluos-IAPP at 5 nM consist mainly of monomers, these results suggested that ACMs bind IAPP monomers/prefibrillar species with high affinity[32]. IAPP/ACM hetero-assemblies were then cross-linked at various incubation time points with glutaraldehyde, separated by NuPAGE, and visualized by Western blot (WB) (Fig. 2b and Supplementary Fig. 7). IAPP alone contained monomers, clearly resolved low MW oligomers (2–6-mers), and fibrils which did not enter the gel (Fig. 2b)[12]. By contrast, a strong smear between ~15 kDa and the upper end of the gel was observed in IAPP/ACM mixtures already at 0 h, indicative of large amounts of medium-to-high MW co-assemblies (Fig. 2b and Supplementary Fig. 7). In addition, bands corresponding to hetero-dimers and hetero-tri-/-tetramers or IAPP mono-/dimers were also present in the IAPP/ACM mixtures, whereas the pattern of

IAPP/non-inhibitor mixtures was as for IAPP alone (Fig. 2b and Supplementary Fig. 7). Formation of hetero-dimers and large MW aggregates was further confirmed by size exclusion chromatography (SEC) (Fig. 2c). In addition, formation of low MW IAPP homo- and IAPP/ACM hetero-oligomers was confirmed by electrospray ionization-ion mobility spectrometry-mass spectrometry (ESI-IMS-MS), a method which had previously been successfully applied to characterize IAPP/Aβ40 hetero-oligomers (Supplementary Fig. 8)[43].

Far-UV-CD spectroscopy revealed that IAPP/Nle3-VF co-assemblies exhibited a mixture of disordered and β-sheet structures (Fig. 2d). By contrast, the CD spectra of aged IAPP and IAPP/non-inhibitor (VGS-VF) mixtures were typical for β-sheet-rich aggregates (Fig. 2d)[44]. Furthermore, anilinonaphthalene 8-sulfonate (ANS) binding studies indicated that IAPP/Nle3-VF co-assembly fully suppressed surface-exposure of hydrophobic clusters, which occurs at early steps of IAPP amyloid self-assembly and is likely related to cytotoxic oligomer formation (Fig. 2e)[2,44].

To characterize the morphology of the IAPP/ACM co-assemblies, solutions used for ThT binding and MTT reduction assays were examined with transmission electron microscopy (TEM). As expected, fibrillar assemblies were major species in aged IAPP and its mixtures with the non-inhibitor VGS-VF (Fig. 2f). However, we were surprised to see that the aged mixtures of IAPP with all six ACMs exclusively consisted of fibrillar assemblies as well; these fibrils were indistinguishable from fIAPP fibrils by TEM (6–10 nm widths and 100–200 nm lengths) (Fig. 2f and Supplementary Table S2). Notably, in contrast to aged IAPP and IAPP/non-inhibitor mixtures, no turbidity, gelation, or precipitation, was observed in the above IAPP/ACM mixtures. X-ray fiber diffraction then revealed that fibrils in IAPP/Nle3-VF mixtures exhibited the cross-β pattern, which is typical for amyloid fibrils (Fig. 2g)[1]. Because fIAPP strongly binds ThT and ACMs were non-amyloidogenic up to at least 100 μM (Fig. 1d, Supplementary Fig. 5a and b), it seemed reasonable to speculate that the fibrils in the IAPP/ACM mixtures could be: (a) fIAPP which escaped detection by the ThT binding assay or (b) fIAPP which was covered with non-specifically bound ACMs; competition of ACMs and ThT for the same fIAPP binding sites might have blocked ThT binding to fIAPP. However, these possibilities were excluded by a series of experiments (Supplementary Fig. 9). Of note, most of the fibrillar assemblies of IAPP/ACM mixtures did not bind the amyloid dye Congo red as well (Supplementary Fig. 10).

The ThT-invisible and non-cytotoxic fibrils found in the IAPP/ACM incubations (termed "hf-IAPP/ACM") might thus be heteromeric. To obtain more evidence for this hypothesis, we first applied immunogold-TEM. In fact, aged IAPP/ACM mixtures contained fibrils which bound both the anti-IAPP and the anti-Aβ (anti-ACM) antibody (Fig. 3a and Supplementary Fig. 11a). Additional support was obtained by hetero-complex pull-down assays. Here, ThT-invisible fibrils present in aged mixtures of N-terminal biotin-labeled IAPP (Biotin-IAPP) with Nle3-VF were captured by streptavidin-coated magnetic beads and their components revealed by WB (Fig. 3b).

High-resolution advanced laser-scanning microscopy provided further unequivocal evidence for diverse supramolecular IAPP/ACM nanofiber co-assemblies (Fig. 3c–i, Supplementary Fig. 11b–f, Supplementary Fig. 12, and Supplementary Movies 1, 2). CLSM, STED, and 2PM visualization of aged IAPP/Nle3-VF mixtures containing N-terminal TAMRA-labeled IAPP (TAMRA-IAPP) and N-terminal Atto647-labeled Nle3-VF (Atto647N-Nle3-VF) or N-terminal fluorescein-labeled Nle3-VF (Fluos-Nle3-VF) revealed large amounts of μm-long heteromeric nanofiber bundles (Fig. 3c, d, Supplementary Fig. 11b–e, and Supplementary Movies 1, 2). Their widths were 232 ± 76 nm ($n = 33$), whereas fIAPP homomeric nanofiber assemblies formed under identical conditions were less broad (124 ± 23 nm ($n = 19$)), as estimated by the more accurate STED nanoscopy (Fig. 3c and Supplementary Fig. 11f). 3D reconstructions of z-stacks of 2PM pictures suggested that heteromeric nanofiber bundles consist of laterally co-assembled, parallel

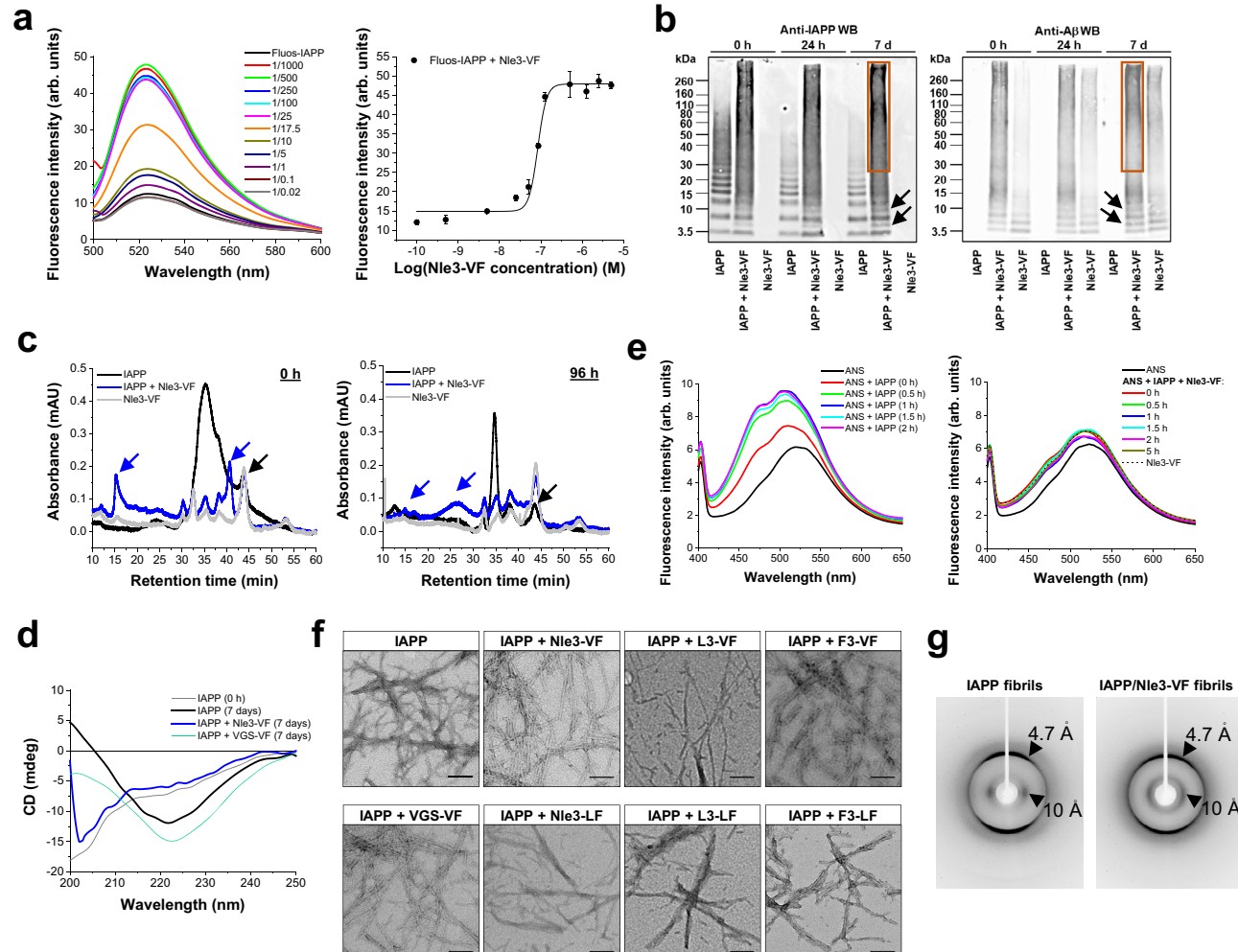

**Fig. 2 | Nanomolar affinity IAPP/ACM interactions yield amyloid fibril-resembling but ThT-invisible nanofibers. a** Nanomolar affinity IAPP/ACM interactions as determined by fluorescence spectroscopy. Fluorescence spectra of Fluos-IAPP (5 nM) and its mixtures with Nle3-VF (pH 7.4) at indicated molar ratios (data from one representative binding assay from *n* = 3 independent assays). Inset, binding curve; data are means ± SD from *n* = 3 independent assays. **b** IAPP/ACM interactions result in hetero-dimers and medium-to-high MW hetero-assemblies. Characterization of IAPP/Nle3-VF hetero-complexes via cross-linking at different time points and NuPAGE and Western blot using anti-IAPP (left) or anti-Aβ (right) antibodies. IAPP/Nle3-VF mixtures (1/2; IAPP, 30 μM) were cross-linked with glutaraldehyde; the orange box indicates medium-to-high MW hetero-assemblies (major species); arrows indicate hetero-di-/-tri-/tetramers. Representative results from >3 independent assays. **c** Characterization of IAPP/Nle3-VF hetero-assemblies via size exclusion chromatography (SEC). Chromatograms of IAPP alone (16.5 μM) or its mixtures with Nle3-VF (1/2) at 0 h and at 96 h (mAU, milli-absorbance units). The black arrow indicates IAPP monomers and Nle3-VF dimers; blue arrows indicate IAPP/Nle3-VF hetero-dimers and medium-to-high MW hetero-assemblies. Similar results were found in two independent assays. **d** IAPP/Nle3-VF co-assemblies are more disordered than β-sheet-rich IAPP assemblies. Far-UV CD spectra of 7-day-

aged IAPP (16.5 μM; pH 7.4) and its mixtures (1/2) with the Nle3-VF or VGS-VF (non-inhibitor) are shown; for comparison, the spectrum of freshly dissolved IAPP (0 h) is also shown. CD spectra are the average of three spectra of the same solution. **e** IAPP/Nle3-VF co-assembly blocks surface-exposure of hydrophobic clusters occurring at early steps of IAPP amyloid self-assembly as determined by anilino-naphthalene 8-sulfonate (ANS) binding. Fluorescence emission spectra of ANS alone/with IAPP (2 μM) (left) and of ANS alone/with IAPP/ACM (1/2) mixtures (right) (pH 7.4) were measured at various time points of self- or co-assembly as indicated (representative results from two independent assays). **f** IAPP/ACM interactions result in ThT-invisible fibrils of indistinguishable appearance to fIAPP by TEM. TEM images of 7-day-aged IAPP (16.5 μM) and its mixtures (1/2) with ACMs or VGS-VF (non-inhibitor) (solutions from Fig. 1d, f). Scale bars: 100 nm. Representative images from seven independent assays for fIAPP and the IAPP/Nle3-VF mixture and two to three similar independent assays for the other IAPP/ACM mixtures. **g** fIAPP and fibrils in IAPP/Nle3-VF mixture exhibit the amyloid cross-β structure signature. X-ray fiber diffraction patterns of fIAPP and fibrils present in aged IAPP/Nle3-VF mixture (1/2) showed major meridional and equatorial reflections at ~4.7 and ~10 Å. Data were representative of two independent experiments.

arranged/in part intertwined stacks of IAPP and ACM molecules (Supplementary Movies 1, 2 and Supplementary Fig. 11d, e). Additional 2PM studies revealed diverse highly ordered fibrous superstructures, including huge macromolecular loops (~500 μm long) (Fig. 3g, h and Supplementary Fig. 12a) and ribbon- or nanotube-like co-assemblies (widths 5–20 μm, lengths >500 μm) (Fig. 3h, i). Interestingly, parts of the ribbon-like co-assemblies were reminiscent of giant DNA double helices (Fig. 3h, i). Here, IAPP assemblies seemed to "wrap" and "link" two parallel-running heteromeric nanofiber bundles and similar observations were made in the nanotube-like co-assemblies. Twisted

heteromeric nanofiber bundles were also observed (Fig. 3h). Notably, IAPP mixtures with other ACMs but not with the non-inhibitor VGS-VF contained similar heteromeric nanofiber superstructures as the IAPP/Nle3-VF mixtures (Supplementary Fig. 12b–d).

At this stage, detailed studies on the interaction of ACMs with fIAPP were performed (Supplementary Fig. 13). Dot blots showed that ACMs and the non-inhibitor VGS-VF bind fIAPP. However, ACM/fIAPP co-assemblies (termed "ACM-coated" fIAPP) consisted of fIAPP bundles which were randomly covered by amorphous Nle3-VF aggregates and maintained the ThT binding and cytotoxic properties of fIAPP

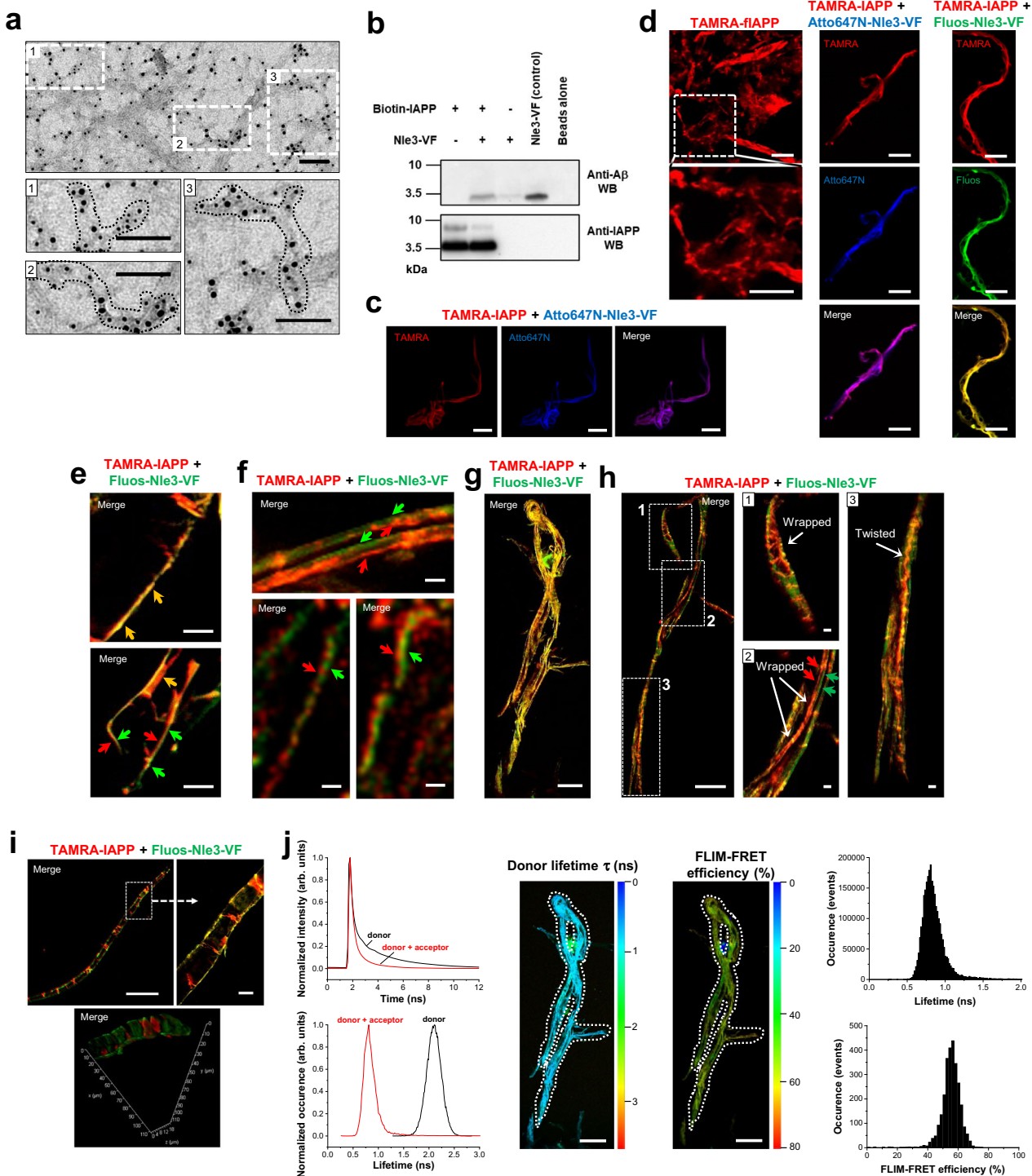

(Supplementary Fig. 13 and Supplementary Movie 3; see also Supplementary Fig. 9). These results further supported the notion that hf-IAPP/ACM were distinct from ACM-coated flAPP.

To learn more about the molecular architecture of the IAPP/ACM nanofibers, we used FLIM-FRET. Pronounced FLIM-FRET events were observed in TAMRA-IAPP/Fluos-Nle3-VF nanofiber co-assemblies (Fig. 3j). The faster donor (Fluos-Nle3-VF) fluorescence decay, its strongly reduced lifetime (~0.8 ns) in presence of the acceptor (TAMRA-IAPP), and the appreciable FLIM-FRET efficiency (~55%) were consistent with a very close donor-acceptor proximity i.e., <5.5 nm corresponding to the Förster radius of the TAMRA/Fluos pair (Fig. 3i and Supplementary Fig. 14). The FLIM-FRET data supported the notion that IAPP and the ACM might be part of the same fibril.

Together, these results suggested that the potent inhibitory effect of ACMs is mediated by nanomolar affinity co-assembly with IAPP monomers/prefibrillar species into amyloid fibril-resembling but ThT-invisible and non-cytotoxic nanofibers and their diverse highly ordered superstructures.

## ACM/IAPP nanofibers evolve from amorphous co-assemblies and IAPP may act as a template

We next asked at which stage of the co-assembly process the fibrillar co-assemblies form. The cross-linking and SEC studies indicated that large hetero-assemblies were present already at the beginning of the co-incubation (Fig. 2b, c). TEM only detected amorphous aggregates between 0 and 48 h, whereas fibrils were the most abundant species at

**Fig. 3 | Evidence for supramolecular IAPP/ACM nanofiber co-assembly.**
**a** Immunogold-TEM images of aged IAPP/Nle3-VF (1/2; IAPP, 16.5 μM) reveals fibrils which bind to both anti-fIAPP and anti-Aβ antibodies (IAPP, 5 nm gold; Nle3-VF, 10 nm gold). Scale bars: 100 nm. Representative images from four biologically independent samples. **b** Heteromeric nature of ThT-invisible fibrils in aged Biotin-IAPP/Nle3-VF (1/2; Biotin-IAPP, 16.5 μM) as assessed by a biotin pull-down assay. Components were revealed by WB with anti-Aβ (upper part) and anti-IAPP (lower part) antibodies; lane "Nle3-VF (control)", Nle3-VF directly loaded onto the gel (without beads) (data representative from two independent assays). **c** STED images of supramolecular heteromeric nanofiber bundles in aged IAPP/Nle3-VF (1/2; IAPP(total), 16.5 μM) containing TAMRA-IAPP and Atto647-Nle3-VF (10%). Scale bars: 5 μm. Nanofiber assemblies are representative of 11 assemblies found in one sample; consistent with the results of 2PM examination of three biologically independent samples (see **d**). **d** 2PM images of nanofiber bundles in aged IAPP-containing TAMRA-IAPP (10%) (left), aged IAPP/Nle3-VF containing TAMRA-IAPP and Atto647N-Nle3-VF (10%) (middle), and aged IAPP/Nle3-VF containing TAMRA-IAPP and Fluos-Nle3-VF (10%) (right) (1/2; IAPP(total), 16.5 μM). Scale bars: 10 μm. Assemblies are representative of assemblies observed in three (IAPP/ACM) and four (IAPP) biologically independent samples examined by 2PM (IAPP/Nle3-VF) and by 2PM or CLSM & STED (IAPP/Nle3-VF), respectively. **e–h** 2PM images of heteromeric fibrous superstructures in aged TAMRA-IAPP/Fluos-Nle3-VF (1/2) (TAMRA-IAPP 16.5 μM). Short colored arrows indicate nanofiber bundles parallel or intertwined (red, TAMRA-IAPP; green, Fluos-Nle3-VF) or overlaying (yellow); long white arrows indicate twists or wrapping. Scale bars: panel (**e**) 5 μm, **f** upper part, 5 μm and lower parts, 1 μm, **g** 50 μm, **h** 50 μm (insets 5 μm). Images are representative of three biologically independent samples. **i** 2PM image of a huge nanotube-like co-assembly found in aged TAMRA-IAPP/Fluos-Nle3-VF (1/2; TAMRA-IAPP, 16.5 μM) (upper panel) and 3D reconstruction of z-stacks (lower panel). Scale bars: 100 μm (inset 10 μm). Data were representative from three biologically independent samples. **j** FLIM-FRET analysis of TAMRA-IAPP/Fluos-Nle3-VF co-assembly of **g** indicates a very close (<5.5 nm) donor-acceptor proximity. Left panel, fluorescence decay curves (top) and lifetimes (bottom) of donor (Fluos-Nle3-VF) without or with acceptor (TAMRA-IAPP); a strong shift of donor lifetime in the presence of acceptor is observed. Middle panel/left side, FLIM image showing donor lifetime; lifetime range 0 ns (dark blue) to 3.5 ns (red); scale bar, 50 μm. Middle panel/right side, FLIM-FRET efficiency (%); efficiency range 0% (dark blue) to 80% (red); scale bar, 50 μm. Right panel, distributions donor lifetime (<1 ns) and FLIM-FRET efficiency (~55%). Data were representative of two independent experiments.

later time points (7 days) (Fig. 4a and Supplementary Fig. 15a). Notably, early amorphous aggregates in the IAPP/ACM mixtures were non-cytotoxic as also found for the fibrils (Supplementary Fig. 15b). Thus, non-toxic hf-IAPP/ACM likely evolve via structural rearrangements of non-toxic amorphous co-aggregates.

Because ACMs were non-amyloidogenic in isolation but co-assembled with IAPP into amyloid fibril-resembling nanofibers, we hypothesized that the amyloidogenic character of IAPP could play a role[13]. In fact, TEM showed that no fibrils formed in mixtures of Nle3-VF with the natively occurring (human) IAPP analog rat IAPP or the earlier designed double N-methylated IAPP analog IAPP-GI, which have high sequence identity to IAPP but are weakly or non-amyloidogenic (Fig. 4b)[3,32]. Moreover, "cross-nucleation" studies using 2PM and FLIM-FRET suggested a templating role for IAPP monomers/prefibrillar species (Fig. 4c, d). Addition of seed amounts (5%) of TAMRA-IAPP monomers to Fluos-Nle3-VF yielded within 48 h nanofiber co-assemblies of similar appearance and FLIM-FRET properties as those present in 6-day-aged TAMRA-IAPP/Fluos-Nle3-VF (1/2) mixtures (Figs. 4c, d, 3f, g, Supplementary Fig. 16, Supplementary Movie 4). Notably, no appreciable FLIM-FRET events were detected when TAMRA-fIAPP seeds were used (Supplementary Fig. 17).

**Additional properties of hf-IAPP/ACM**
We next studied whether non-toxic hf-IAPP/ACM may differ from fIAPP regarding other properties as well. First, we asked whether hf-IAPP/ACM can seed IAPP fibrillogenesis[44]. However, in contrast to fIAPP or fibrils present in IAPP/non-inhibitor mixtures, seed amounts of hf-IAPP/ACM were unable to accelerate IAPP fibrillogenesis as assessed by ThT binding (Fig. 4e).

Pathogenic amyloid fibrils are usually characterized by an extraordinarily high stability[1,45]. Therefore, we compared the thermostabilities of IAPP/ACM nanofibers and fIAPP by using ThT binding and TEM. In contrast to fIAPP, hf-IAPP/Nle3-VF were fully converted into amorphous aggregates after heating to 95 °C for 5 min (Fig. 4f). Furthermore, most pathogenic amyloids are resistant toward proteolytic degradation, including PK degradation[45]. fIAPP and hf-IAPP/ACM were therefore incubated with PK and degradation kinetics followed by dot blot analysis using anti-fIAPP and anti-Aβ (anti-ACM) antibodies (Fig. 4g). Remarkably and in contrast to fIAPP, which was stable to PK digestion for at least 30 h, hf-IAPP/ACM were fully degraded in <6 h with their ACM component degraded within few minutes. Finally, we studied the cellular uptake efficiency of hf-IAPP/ACM in direct comparison to fIAPP. In fact, the uptake of amyloid assemblies by macrophages and microglia is a major mechanism of amyloid clearance in both AD or T2D[46–48]. Uptake of fIAPP versus hf-IAPP/ACM was studied

in primary murine bone marrow-derived macrophages (BMDMs) and the well-established murine microglial cell line BV2 by fluorescence microscopy using TAMRA-IAPP as tracer[47–50]. We found that three- to ten-fold higher amounts of hf-TAMRA-IAPP/ACM were phagocytosed by both cell types as compared to TAMRA-fIAPP (Fig. 4h and Supplementary Fig. 18).

Together, the results revealed several potentially beneficial properties of non-toxic hf-IAPP/ACM, i.e., seeding incompetence, thermolability, high PK sensitivity, and an efficient cellular clearance profile, which clearly distinguished them from pathogenic fIAPP.

**Proposed hypothetical models of IAPP/ACM nanofiber co-assembly**
The structure of Aβ/IAPP hetero-amyloids has not yet been elucidated[18]. Based on the suggested structures of fIAPP and fAβ40(42) and the polymorphic nature of the self-/cross-amyloid assembly, various different interfaces could be involved in hf-IAPP/ACM formation (Fig. 5)[18,24,25,27–31,51,52]. Our data suggested that single IAPP/ACM nanofibers and basic units of their superstructures may form by lateral co-assembly of two or more "protofilament"-like stacks of IAPP and ACM molecules (Fig. 5). Thereby, β-sheet-prone segments of the ACM part corresponding to Aβ(21–40) would become incorporated into the β-sheet H-bond network of the ACM "protofilament" yielding diverse cross-interaction surfaces with both IAPP and ACM stacks (Fig. 5a, b)[18,25,27–29]. The proposed arrangement of Aβ(21–40) is also supported by the finding that Aβ(15–40) analogs with N-methylations within Aβ(21–40) did not inhibit (Supplementary Fig. 2a–c). IAPP "protofilament" cross-interactions with the ACM could be mediated via previously suggested IAPP segments and fIAPP folds or yet unknown variants thereof[24,25,27,30]. The lack of a second IAPP "protofilament" and a non-ideal IAPP/Aβ(ACM) side chain interdigitation in the hetero-nanofiber as compared to fIAPP and/or intrinsic instability of involved IAPP and/or ACM folds could account for hetero-nanofiber lability[45,53]. Notably, axial IAPP/ACM "protofilament" co-assembly could also occur (Fig. 5c)[28]. However, the STED and 2PM data, the expected higher instability of such a "protofilament" due to the N-methylations, and the observed lack of inhibitory effects of partial ACM segments (Supplementary Fig. 2f–h) make this scenario less likely.

**ACMs inhibit Aβ42 amyloid self-assembly via co-assembly into ThT-invisible and non-toxic nanofibers and their diverse superstructures**
In analogy to other Aβ-derived Aβ inhibitors, ACMs would be expected to also interfere with Aβ amyloid self-assembly[19,33,40,41]. In fact, ThT

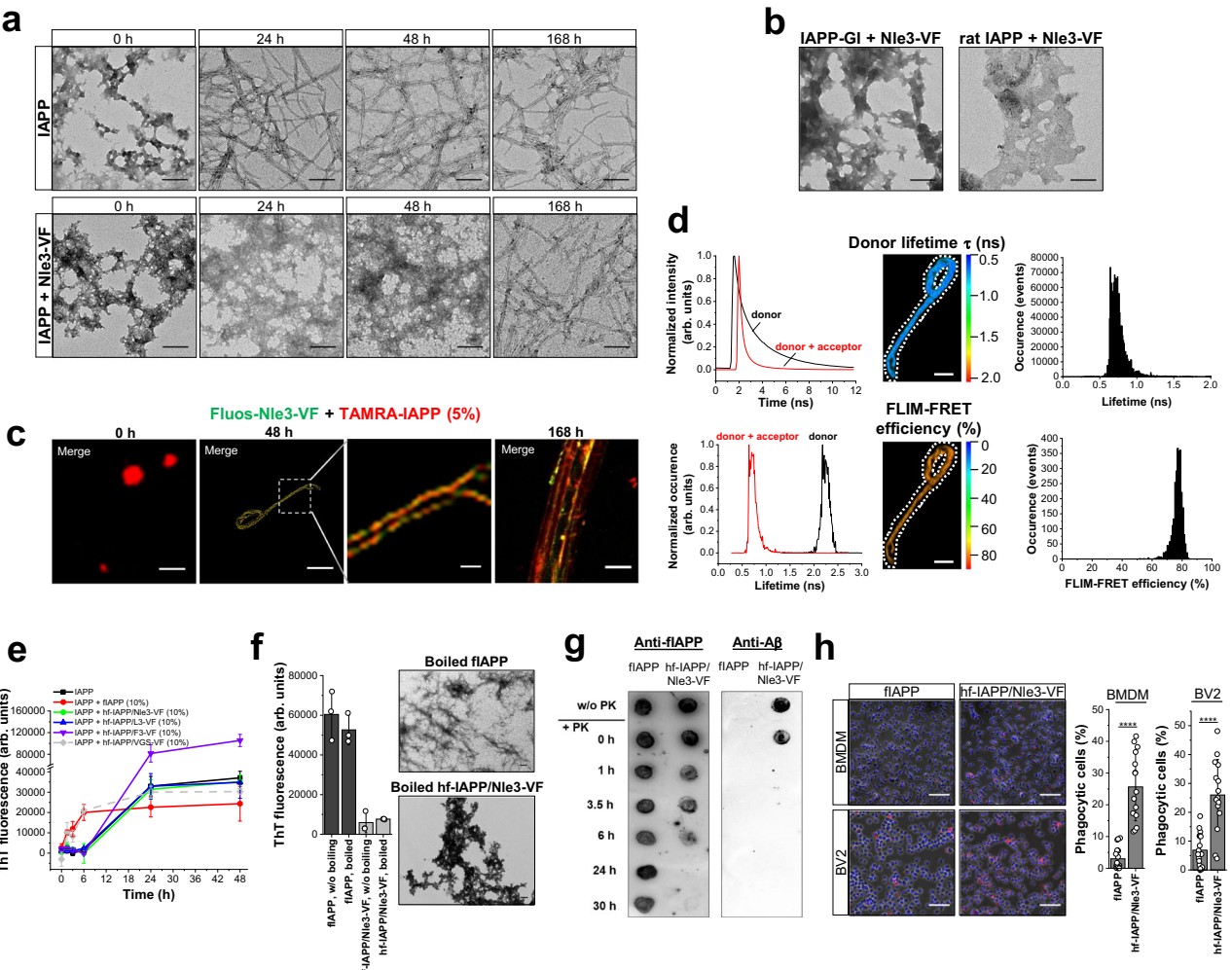

**Fig. 4 | Mechanism of formation and properties of IAPP/ACM nanofiber co-assemblies. a** Evolution of hf-IAPP/ACM from amorphous co-aggregates. TEM images of IAPP (16.5 μM) and IAPP/Nle3-VF mixtures (1/2) between 0 and 7 days of incubation. Scale bars: 100 nm. Data were representative from two independent similar experiments. **b** TEM images of 7-day-aged IAPP-GI/Nle3-VF or rat IAPP/Nle3-VF (1/2) show amorphous aggregates. Images show major aggregate populations present in each sample. Consistent results were found by 2PM for IAPP-GI/Nle3-VF. Scale bars: 100 nm. **c** IAPP monomers/prefibrillar species template nanofiber co-assembly. Representative 2PM images of Fluos-Nle3-VF (33 μM) cross-seeded with freshly made TAMRA-IAPP (5%). Scale bars: 10 μm; inset, 1 μm. Images are from one sample examined at various incubation time points and data are consistent with 2PM data of Fig. 3e–j. **d** FLIM-FRET of nanofiber co-assembly of **c** at 48 h reveals similar features to hf-TAMRA-IAPP/Fluos-Nle3-VF (1/2; 6-day-aged) from Fig. 3j. Left panel, fluorescence decay curves (top) and lifetimes (bottom) of Fluos-Nle3-VF without or with TAMRA-IAPP shows a strong shift of donor lifetime in the presence of acceptor. Middle panel/upper part, FLIM image showing donor lifetime; range as indicated; scale bar, 5 μm. Middle panel/lower part, FLIM-FRET efficiency (%); range as indicated; scale bar, 5 μm. Right panel, distributions donor lifetime (<1 ns) and FLIM-FRET efficiency (~75%). Data from one experiment. **e** hf-IAPP/ACM are seeding-incompetent. IAPP (12 μM) fibrillogenesis alone or with 10% hf-IAPP/ACM,

flAPP, or IAPP/VGS-VF was followed by ThT binding (means ± SD from $n = 4$ (IAPP alone and IAPP with 10% flAPP) or $n = 3$ (all other samples) independent assays). **f** Thermostability of hf-IAPP/ACM versus flAPP. Left panel, ThT binding of flAPP and hf-IAPP/Nle3-VF before/after boiling (5 min); means ± SD, three independent assays. Right panel, representative TEM images after boiling; scale bars: 100 nm. Results are representative from two similar independent experiments. **g** Degradation of hf-IAPP/ACM versus flAPP by proteinase K (PK) followed by dot blot. flAPP or hf-IAPP/Nle3-VF were subjected to PK digestion (37 °C); quantification by anti-flAPP and anti-Aβ antibodies. Representative membranes from three independent assays. **h** Phagocytosis of hf-IAPP/ACMs versus flAPP by primary murine BMDMs and cultured murine BV2 microglia. Left panel, representative microscopic images of cells following incubation (6 h, 37 °C) with TAMRA-flAPP (3.3 μM) or hf-TAMRA-IAPP/Nle3-VF (3.3 μM); red dots, TAMRA-IAPP; scale bars, 100 μm. Mid and right panels, amounts of phagocytic cells (% of total). Data means ± SD from 18 or 15 biologically independent samples of TAMRA-flAPP or hf-TAMRA-IAPP/Nle3-VF, respectively, analyzed in five independent cell assays with each assay well analyzed in three fields of view. ****$P < 0.0001$ for hf-TAMRA-IAPP/Nle3-VF versus TAMRA-flAPP, i.e., $P = 1.2349E-09$ in BMDM cells and $1.5781E-06$ in BV2 cells (unpaired $t$-test (two-sided)).

binding and MTT reduction assays in PC12 cells showed that all six ACMs (Aβ42/ACM 1/1) effectively suppressed the formation of Aβ42 fibrils and cytotoxic assemblies (Fig. 6a, b). Titrations of cytotoxic Aβ42 with ACMs yielded mostly nanomolar $IC_{50}$ values consistent with potent inhibitory activity (Table 2 and Supplementary Fig. 19). Furthermore, ACMs strongly suppressed seeding of Aβ42 fibrillogenesis by preformed fAβ42 (Fig. 6c).

Remarkably, TEM examination of the aged "ThT-negative" and non-cytotoxic Aβ42/ACM mixtures revealed that they exclusively

consisted of fibrils (Fig. 6d). These fibrils were 2–4 times longer than fAβ42 while their widths were identical to the widths of fAβ42 (7–8 nm) (Fig. 6d and Supplementary Table 3). Additional TEM and ThT studies suggested that the formation of long fibrils was linked to inhibitory activity (Supplementary Fig. 20). Of note, most of the fibrillar assemblies in Aβ42/ACM mixtures were non-birefringent under polarized light when stained with CR (Supplementary Fig. 21). Because the fibrils in the Aβ42/ACM mixtures were significantly longer than fAβ42, non-toxic, and could not solely consist of fAβ42, which binds both ThT and

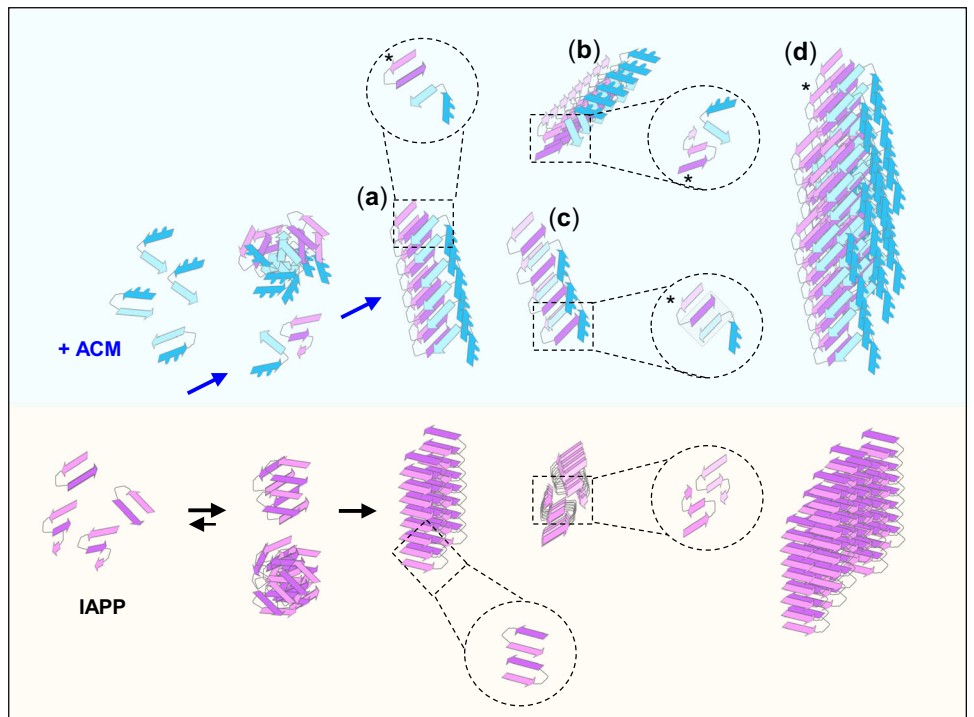

**Fig. 5 | Proposed mechanism and hypothetical models of IAPP/ACM nanofiber co-assembly versus IAPP amyloid self-assembly.** The lower part, IAPP self-assembly into toxic oligomers and amyloid fibrils. The upper part, in the presence of ACMs, IAPP monomers/prefibrillar species are redirected into initially amorphous and non-toxic hetero-assemblies, which convert into amyloid fibril-resembling but ThT-invisible and non-toxic heteromeric nanofibers and their fibrous superstructures. Shown are hypothetical models of heteromeric nanofibers (**a**–**c**) and supramolecular co-assemblies thereof (**d**) generated by lateral (**a**, **b**, **d**) or axial (**c**) co-assembly of the ACM with two of the previously suggested fIAPP folds or variants thereof (indicated by "*")[28,30]. The ACM is shown in Aβ amyloid core-mimicking strand-loop-strand folds[31]; blue dots indicate *N*-methyl rests.

CR, or the non-amyloidogenic ACMs, we assumed that they might be heteromeric (termed hf-Aβ42/ACM).

2PM examination of aged Aβ42/ACM mixtures containing N-terminal TAMRA-labeled Aβ42 (TAMRA-Aβ42) and Fluos-ACMs revealed diverse heteromeric fibrous superstructures. These comprised several μm-long heteromeric nanofiber bundles with widths between 0.5–2 μm and related heterogeneous superstructures, i.e. ribbons, tapes, or nanotube-like ones with widths between 3–14 μm (Fig. 6e and Supplementary Fig. 22). The 2PM images and 3D reconstructions of z-stacks indicated that both axial and lateral co-assembly might underlie their formation, likely enabled by the high degree of sequence identity between ACMs and Aβ(15–40) (Supplementary Movie 5). In hf-Aβ42/Nle3-VF, we observed thick "nodes" periodically arranged along a long "cable"-like part (Fig. 6e and Supplementary Movie 5). Pronounced FLIM-FRET events were detected, which were consistent with close distances between the two peptides, i.e., <5.5 nm (Fig. 6f and Supplementary Fig. 23). Notably, a stronger reduction of Fluos-Nle3-VF lifetime in the presence of TAMRA-Aβ42, i.e., from ~2.1 to ~0.8 ns, and higher FLIM-FRET efficiency (~60%) were observed in the node regions of interest (ROI-1) than in the "cable"-like ones (ROI-2) consistent with their structural heterogeneity (Fig. 6f).

ACM/Aβ42 interactions and hetero-complexes were then studied by fluorescence spectroscopy, SEC, cross-linking in combination with NuPAGE and WB, and far-UV CD spectroscopy (Supplementary Fig. 24). Fluorescence spectroscopic titrations of N-terminal FITC-labeled Aβ42 (FITC-Aβ42, 5 nM) with ACMs yielded nanomolar app. $K_D$s (Table 2 and Supplementary Fig. 24a). SEC and cross-linking studies revealed large MW hetero-assemblies in Aβ42/Nle3-VF mixtures consistent with the TEM and 2PM findings (Supplementary Fig. 24b, c). Cross-linking also identified Aβ42/Nle3-VF hetero-dimers; their formation might underlie hetero-nanofiber co-assembly. Far-UV CD spectroscopy indicated less β-sheet structure in hf-Aβ42/Nle3-VF as compared to fAβ42 (Supplementary Fig. 24d).

Together, these results suggested that the potent inhibitory effect of ACMs on Aβ42 amyloid self-assembly is mediated by nanomolar affinity interactions of ACMs with Aβ42 monomers/prefibrillar species, which redirect them into long ThT-invisible and non-toxic hetero-nanofibers and their diverse μm-scaled superstructures. Further studies suggested that the binding of ACMs to preformed fAβ42 does not result in this kind of co-assemblies (Supplementary Fig. 25).

### Additional properties of Aβ42/ACM co-assemblies

Hippocampal synaptic plasticity is regarded as a key mediator of learning and memory processes; its damage by toxic Aβ42 aggregates is a major responsible factor in AD pathogenesis[54,55]. Our ex vivo electrophysiological studies in mouse brains revealed that in the presence of various different ACMs, Aβ42-mediated inhibition of hippocampal long-term potentiation (LTP) was fully ameliorated (Fig. 7a). Of note, ACMs exerted their inhibitory effects when added to both Aβ42 monomers and pre-oligomerized Aβ42 (Fig. 7a and Supplementary Fig. 26). As inhibition of LTP by Aβ42 is linked to loss of memory and cognitive functions in AD, this data supported the potential physiological relevance of the in vitro determined inhibitory effects[54,55].

We then investigated whether hf-Aβ42/ACM might differ from fAβ42 with respect to their seeding competence. In fact, ThT binding showed that, in contrast to fAβ42, hf-Aβ42/Nle3-VF, and hf-Aβ42/L3-VF were seeding-incompetent (Fig. 7b). Furthermore, we asked whether hf-Aβ42/ACM might exhibit similar proteolytic degradation, thermolability, and cellular clearance features as hf-IAPP/ACM. Kinetics of PK-mediated degradation of fAβ42 versus hf-Aβ42/Nle3-VF were studied by dot blot analysis and the 6E10 antibody, which specifically recognizes Aβ42 (Aβ(1–17)) but not the ACMs. hf-Aβ42/Nle3-VF were

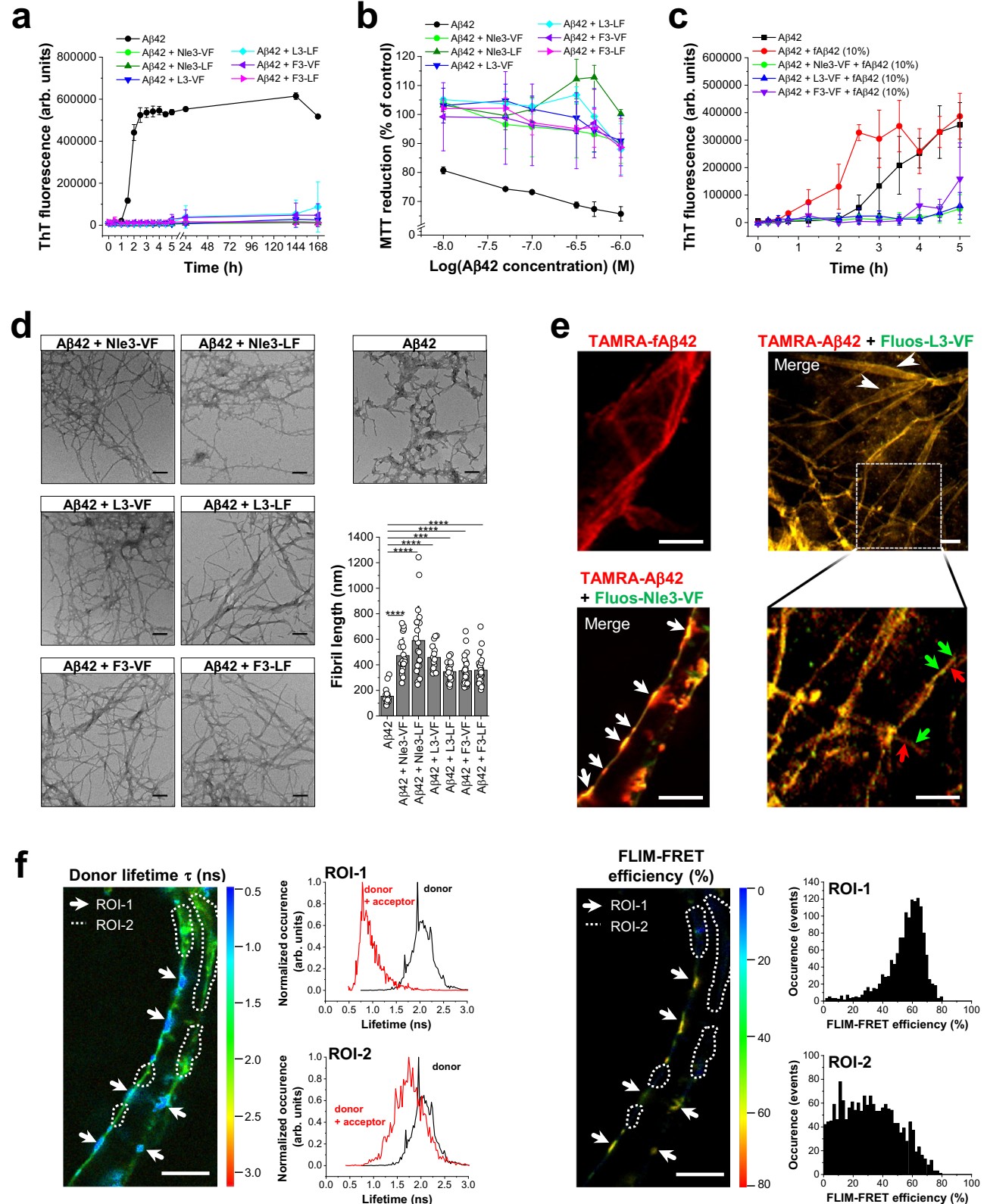

degraded within ~30 min, whereas degradation of fAβ42 took ~2 h, i.e., was ~4 times slower (Fig. 7c). Thermostability studies using TEM then showed that hf-Aβ42/Nle3-VF were fully converted into amorphous aggregates after 5 min at 95 °C; by contrast, fAβ42 were stable at 95 °C for at least 15 min (Fig. 7d). Finally, phagocytosis of TAMRA-fAβ42 and hf-TAMRA-Aβ42/Nle3-VF(L3-VF) by cultured BV2 microglia cells was quantified by fluorescence microscopy[46,48–50]. Significantly higher

amounts of hf-TAMRA-Aβ42/ACM became phagocytosed as compared to TAMRA-fAβ42 (Fig. 7e). Of note, ACM addition to preformed cyto-toxic TAMRA-Aβ42 oligomers (TAMRA-oAβ42) resulted in a delay of fibrillogenesis, less cytotoxicity, and a significantly increased cellular uptake (Supplementary Fig. 27).

Together, these findings revealed that ThT-invisible and non-toxic hf-Aβ42/ACM were seeding-incompetent and less thermostable than

**Fig. 6 | Inhibition of Aβ42 amyloid self-assembly via ThT-invisible and non-toxic Aβ42/ACM nanofiber co-assembly. a** ACMs inhibit Aβ42 amyloid self-assembly. Fibrillogenesis of Aβ42 (5 μM) and Aβ42/ACM (1/1) followed by ThT binding (means ± SD, three independent assays). **b** ACMs suppress Aβ42 cytotoxicity. Aged Aβ42 (5 μM) or Aβ42/ACM (1/1) (6 days) (without ThT) were added to PC12 cells; cell damage determined via MTT reduction (means ± SD, three independent assays, $n = 3$ technical replicates each). **c** ACMs suppressed seeding of Aβ42 by fAβ42. Fibrillogenesis of Aβ42 (5 μM) without or with fAβ42 seeds (10%) and of fAβ42-seeded Aβ42/ACM (1/1) followed by ThT binding (means ± SD, three independent assays). **d** Aged Aβ42/ACM consists of ThT-invisible fibrils (hf-Aβ42/ACMs). Representative TEM images of Aβ42 (fAβ42) and Aβ42/ACM mixtures (1/1) (from **b**; 6-day-aged) are shown; scale bars, 100 nm. Images represent results from eight (Aβ42), three (Aβ42/Nle3-VF), two (Aβ42/L3-VF, Aβ42/F3-VF, Aβ42/F3-LF), or one (Aβ42/Nle3-LF, Aβ42/L3-LF) independent experiment(s). Bottom right, bar diagram showing fibril lengths; data from $n = 22$ fibrils in fAβ42 and $n = 20, 20, 15, 20, 23$, and 22 fibrils in Aβ42 mixtures with Nle3-VF, Nle3-LF, L3-VF, L3-LF, F3-VF, and F3-LF, respectively. ***$P < 0.001$ or ****$P < 0.0001$ for lengths of hf-Aβ42/ACMs versus fAβ42 as indicated (one-way ANOVA & Bonferroni). $P$ values: 3.36E-10, 7.41E-17, 3.40E-08, 3.71E-04, 9.25E-05, and 7.52E-05 for Aβ42/Nle3-VF, Nle3-LF, L3-VF, L3-LF, F3-VF, and F3-LF, respectively. **e** 2PM images of fAβ42 and hf-Aβ42/ACMs.

Fibrillar co-assemblies in aged Aβ42-containing TAMRA-Aβ42 (50%) (2 h; fibrillogenesis plateau), aged Aβ42/Fluos-Nle3-VF containing TAMRA-Aβ42/Fluos-Nle3-VF (50%) (4 days), and aged Aβ42/Fluos-L3-VF containing TAMRA-Aβ42/Fluos-L3-VF (50%) (6 days). White arrowheads indicate ribbon- or nanotube-like co-assemblies (yellow); white arrows indicate large "node"-like parts (yellow); colored arrows indicate TAMRA-Aβ42 (red) and Fluos-L3-VF (green) "building units"; scale bars, 10 μm (see also Supplementary Movie 5). Similar findings in 3 (Aβ42 and Aβ42/Nle3-VF) or 2 (Aβ42/L3-VF) biologically independent samples. **f** FLIM-FRET of hf-TAMRA-Aβ42/Fluos-Nle3-VF of **e** indicates regions of high proximity (<5.5 nm) of the two polypeptides. Left panel/left side, FLIM image showing Fluos-Nle3-VF lifetimes in the two regions of interest (ROIs); lifetime range, 0.5 ns (dark blue) to 3 ns (red); scale bar, 10 μm. White arrows indicate ROI-1 (node-like) while dotted lines indicate ROI-2 (cable-like). Left panel/right side, diagrams showing lifetimes of the donor without or with acceptor in ROI-1 or ROI-2; a pronounced reduction of donor fluorescence lifetime in the presence of acceptor is observed; the shift is stronger in ROI-1. Right panel/left side, distribution of FLIM-FRET efficiency (%); efficiency range 0% (dark blue) to 80% (red); scale bar, 10 μm. Right panel/right side, bar diagrams showing FLIM-FRET efficiency (%) distribution in ROI-1 (~60%) and ROI-2 (0–70% with a broad maximum at 20–40%). Consistent findings in two similar independent experiments.

fAβ42 and that they became more efficiently degraded by PK and phagocytosed by BV2 microglia than fAβ42.

## Inhibition of fIAPP-mediated cross-seeding of Aβ42 amyloid self-assembly by ACMs

Cross-seeding of Aβ42 amyloid self-assembly by fIAPP accelerates Aβ42 amyloid self-assembly and could link the onset and pathogenesis of T2D with AD[8,11,14]. In a simplified mechanistic scenario, fIAPP seeds will template the formation of IAPP/Aβ42 hetero-amyloids, which will template further cytotoxic Aβ42 self-/cross-assembly events[1,13]. Thereby, polymorphic cross-interactions between amyloid core regions may play an important role[24–29].

Because ACMs contains the Aβ amyloid core, bind with high affinity to both IAPP and Aβ42, incl. fIAPP and fAβ42, and inhibit their amyloid self-assembly, we assumed that they might also interfere with the cross-seeding of Aβ42 by fIAPP. In fact, ACMs effectively suppressed fIAPP-mediated cross-seeding of Aβ42 fibrillogenesis and cytotoxicity (Fig. 7f). Involved supramolecular (co-)assemblies were then studied by 2PM (Fig. 7g–j). First, preformed TAMRA-fIAPP seeds were added to Aβ42-containing N-terminal HiLyte647-labeled Aβ42 (HiLyte647-Aβ42). 2PM at the fibrillogenesis plateau revealed large Aβ42 clusters, consisting of apparently amorphous aggregates or fibrils, bound to/branching out from fIAPP surfaces (Fig. 7g, h and Supplementary Movies 6, 7). This data was consistent with secondary (cross-)nucleation[56]. However, a completely different picture was obtained when TAMRA-fIAPP seeds were added to a mixture of Aβ42 with Nle3-VF containing HiLyte647-Aβ42 and Fluos-Nle3-VF (Fig. 7i, j). Major species were: (a) large fibrous Aβ42/Nle3-VF/IAPP co-assemblies (many μm long; widths ~1–5 μm) and (b) diverse roundish/elliptical Aβ42/Nle3-VF/IAPP or Aβ42/Nle3-VF co-assemblies (up to ~10 μm) (Fig. 7h–j). Analysis of 2PM images and 3D-reconstructions suggested that fibrous co-assemblies consisted of Aβ42, Nle3-VF, and Aβ42/Nle3-VF bound to fIAPP bundles (Fig. 7i, j and Supplementary Movies 8, 9). Thus, suppression of Aβ42 cross-seeding likely occurs via a dual mechanism (Fig. 8): (1) sequestration of Aβ42 into non-toxic ACM/Aβ42 co-assemblies (both fibrillar and amorphous ones) and (2) binding of non-toxic ACM and ACM/Aβ42 co-assemblies to fIAPP yielding cross-seeding-incompetent and non-toxic ternary nanofiber co-assemblies.

## Discussion

Here we exploited Aβ/IAPP cross-interactions to design Aβ amyloid core mimics (ACMs) as inhibitors of amyloid self-assembly of both IAPP and Aβ42. Collectively, we identified six 26-residue peptides as effective amyloid inhibitors of both IAPP and Aβ42. All six ACMs bound IAPP with nanomolar affinity and blocked its cytotoxic amyloid self-assembly with nanomolar $IC_{50}$ values. In addition, all six ACMs bound Aβ42 with nanomolar affinity and blocked its cytotoxic self-assembly, three of them with nanomolar $IC_{50}$ values. Moreover, ex vivo electrophysiology in murine brains showed a full amelioration of Aβ42-mediated damage of synaptic plasticity by ACMs. Importantly, ACMs also inhibited reciprocal cross-seeding of IAPP and Aβ42 amyloid self-assembly[8,9,11]. ACMs thus belong to the most effective inhibitors of in vitro amyloid self-assembly of IAPP, Aβ42, or both polypeptides[1,19].

Our most remarkable finding was that ACMs, which were non-amyloidogenic in isolation, exerted their potent amyloid inhibitor function via an unexpected mechanism, i.e. by co-assembling with IAPP or Aβ42 into amyloid fibril-resembling but ThT-invisible and non-toxic nanofibers and their diverse highly ordered fibrous superstructures. The latter ones comprised large heteromeric nanofiber bundles and several μm-sized loops, ribbons, and nanotube-like superstructures. Furthermore, non-toxic ternary fibrous co-assemblies consisting of fIAPP, Aβ42, and ACM formed when Aβ42 cross-seeding by fIAPP was performed in the presence of ACMs.

Although unexpected, IAPP(Aβ42)/ACM nanofiber co-assembly was in line with the ACM design concept. In fact, ACMs contained all three Aβ hot segments required for high-affinity interactions with IAPP, Aβ42, and themselves, and combined inbuilt β-sheet extension blocking (N-methylations) with β-sheet stabilization/extension enabling (LTS and Aβ(21–40)) elements[20,24–26,40,42].

**Table 2 | $IC_{50}$ values of inhibitory effects of ACMs on Aβ42-mediated cell damage[a] and app. $K_D$s of ACM/Aβ42 interactions[b][c]**

| ACM | $IC_{50}$ (±SD) (nM)[a] | app. $K_D$ (±SD) (nM)[b][c] |
|---|---|---|
| Nle3-VF | 367 (±79) | 14.5 (±8.0) |
| Nle3-LF | n.d. | 11.1 (±6.0) |
| L3-VF | 261 (±140) | 38.0 (±2.0) |
| L3-LF | n.d. | 2.6 (±1.4) |
| F3-VF | 1032 (±297) | 160.8 (±12.9) |
| F3-LF | 262 (±115) | 430.6 (±7.1) |

[a]$IC_{50}$ values, means (±SD) from three independent titration assays ($n = 3$ technical replicates each); n.d. not determined.
[b]Determined by titrations of N-terminal FITC-labeled Aβ42 (5 nM; pH 7.4) with ACMs.
[c]App. $K_D$s, means (±SD) from three binding curves derived from three independent assays.

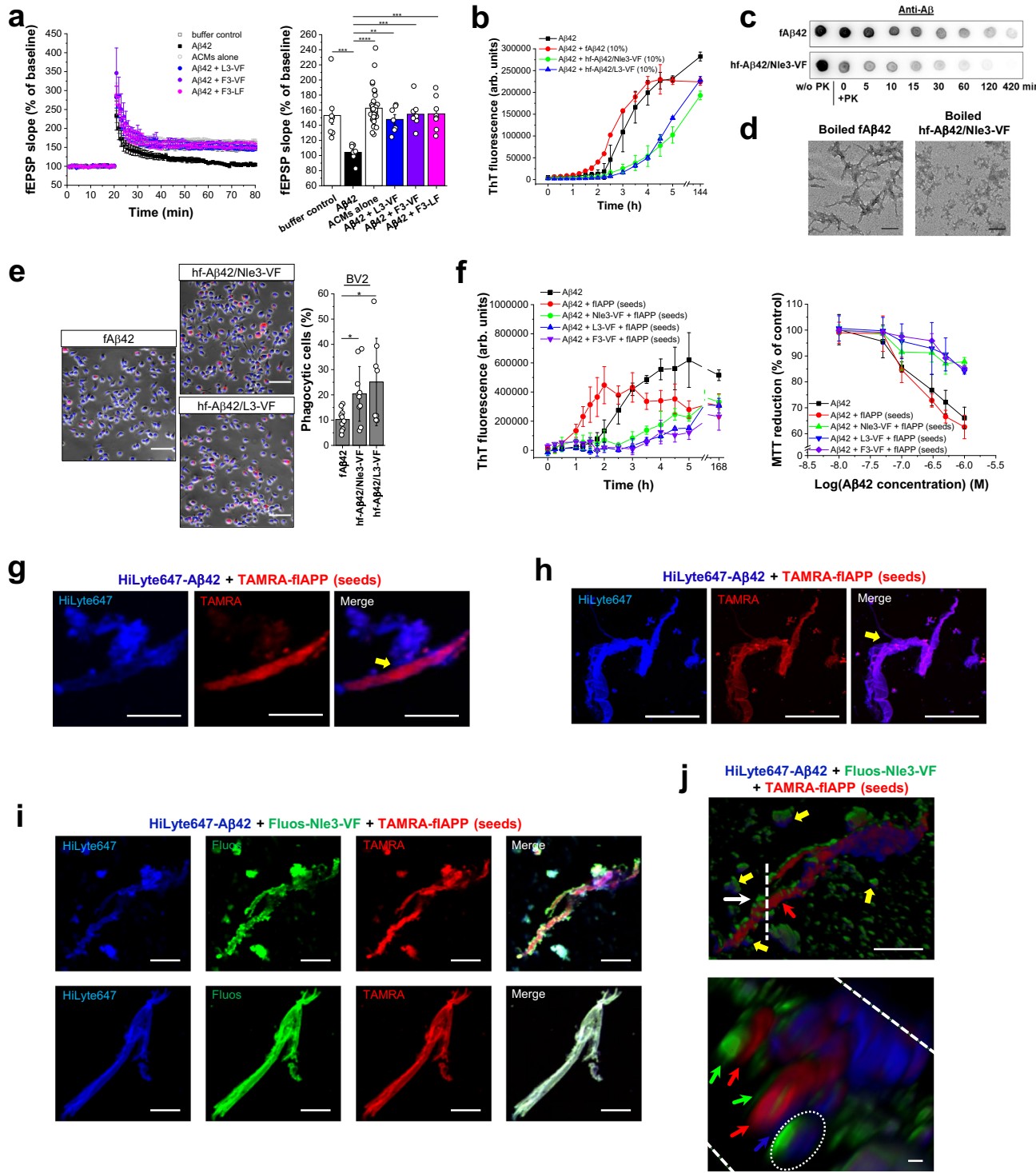

The identified IAPP/ACM nanofibers were indistinguishable from flAPP by TEM and had the cross-β amyloid core signature by XRD, but were less ordered than flAPP according to CD spectroscopy. Our TEM, STED, 2PM, and FLIM-FRET studies suggested that nanofibers and basic parts of their fibrous superstructures consisted of laterally co-assembled, parallel arranged or intertwined/twisted, "protofilament-like" IAPP and ACM stacks. In addition, our studies showed that hf-IAPP/ ACM evolve from large amorphous co-assemblies and suggested that IAPP monomers/prefibrillar species might template this process likely via hetero-dimers. Although the mechanistic steps are yet unclear, IAPP/ACM nanofiber co-assembly could proceed in analogy to proposed mechanisms of self- and co-assembly of IAPP and Aβ[8,12,13,24,29,56].

The identified Aβ42/ACM nanofibers had similar widths but two to four times greater lengths than fAβ42. Our imaging results were consistent with both axial and lateral co-assembly, the former likely underlying hetero-nanofiber elongation. Notably, Aβ42/ACM nanofiber co-assembly was in line with earlier findings by Mihara and coworkers[57].

Another notable finding of our study relates to the potentially beneficial properties of IAPP/ACM and Aβ42/ACM nanofibers. Their properties clearly distinguished from pathogenic flAPP and fAβ42 and, in addition to their non-toxic nature and seeding incompetence, comprised thermolability, proteolytic degradability, and more efficient phagocytosis than flAPP and fAβ42. Such features are reminiscent of labile/reversible functional amyloids, in which they may serve

**Fig. 7 | Properties and functions of Aβ42/ACM co-assemblies. a** Aβ42/ACM co-assembly ameliorates Aβ42-mediated LTP impairment in murine hippocampal slices ex vivo. Left, time course of synaptic transmission ((fEPSP, field excitatory postsynaptic potential). Data were means ± SEM from biologically independent samples as specified: $n = 8$ for Aβ42/F3-VF and Aβ42/F3-LF (1/10), $n = 7$ for Aβ42/L3-VF (1/10)), $n = 8$ for Aβ42 (50 nM) and buffer controls, and $n = 36$ for ACMs alone (500 nM). Right, LTP values: averages from the last 10 min of recording; data, means ± SEM ($n$, see above). **$P < 0.01$, ***$P < 0.001$, and ****$P < 0.0001$ for Aβ42/ACM mixtures versus Aβ42 (one-way ANOVA & Bonferroni) as indicated. $P$ values: 6.90E-04 (buffer versus Aβ42); 8.04E-08 (Aβ42 versus ACMs); 0.0054 (Aβ42 versus Aβ42/L3-VF); 3.99E-04 (Aβ42 versus Aβ42/F3-VF); 3.40E-04 (Aβ42 versus Aβ42/F3-LF). **b** hf-Aβ42/ACM are seeding-incompetent. Aβ42 (5 μM) fibrillogenesis alone or seeded with fAβ42, hf-Aβ42/Nle3-VF, or hf-Aβ42-L3-VF (10%) determined by ThT binding (means ± SD, three independent assays). **c** Degradation of hf-Aβ42/Nle3-VF and fAβ42 by PK (37 °C) followed by dot blot; Aβ42 quantification by Aβ(1–17)-specific antibody. Representative membranes from 3 independent assays. **d** Thermolability of hf-Aβ42/ACM versus fAβ42. Representative TEM images of boiled fAβ42 (15 min) versus hf-Aβ42/Nle3-VF (5 min) (from two independent assays); scale bars: 100 nm. **e** Phagocytosis of hf-Aβ42/ACM versus fAβ42 by cultured murine BV2 microglia. Left and mid panels, representative microscopic images of cells after incubation (6 h, 37 °C) with TAMRA-fAβ42, hf-TAMRA-Aβ42/Nle3-VF, and hf-TAMRA-Aβ42/L3-VF (1 μM); red dots indicate TAMRA-Aβ42; scale bars, 100 μm. Right panel, amounts of phagocytic cells (% of total). Data were means ± SD from 10 (TAMRA-fAβ42 and hf-TAMRA-Aβ42/Nle3-VF) or 8 (hf-TAMRA-Aβ42/L3-VF) biologically independent samples analyzed in two independent cell assays, each assay well analyzed in three fields of view. *$P < 0.05$ as indicated (unpaired $t$-test (two-sided)); $P$ values: 0.0128 and 0. 0179 for hf-Aβ42/Nle3-VF and hf-Aβ42/L3-VF, respectively versus fAβ42. **f** Effects of ACMs on fIAPP-mediated cross-seeding of Aβ42 fibrillogenesis (left panel) or cytotoxicity (right panel). Left panel, fibrillogenesis of Aβ42 (10 μM) or Aβ42/ACM (1/2) mixtures following cross-seeding with fIAPP (20%) and of Aβ42 without fIAPP seeds (10 μM) determined by ThT binding; means ± SD from $n = 8$ (Aβ42 and cross-seeded Aβ42) and $n = 4$ (cross-seeded Aβ42/ACM mixtures) independent assays. Right panel solutions (made as for left panel without ThT; 1.5 h-aged) were added to PC12 cells; cell damage was determined via MTT reduction (means ± SD, three independent assays, $n = 3$ technical replicates each). **g–j** 2PM characterization of supramolecular co-assemblies in Aβ42 solutions after cross-seeding with fIAPP (20%) in the absence (**g, h**) or presence of ACM (**i, j**). **g, h** 2PM images of TAMRA-fIAPP-cross-seeded Aβ42-containing HiLyte647-Aβ42 (50%) (1.5 h; incubations as in **f**) show clusters of Aβ42 assemblies bound to/branching out of fIAPP surfaces; yellow arrow, Aβ42-fIAPP "contact site"; scale bars: 10 μm (**g**) and 100 μm (**h**) (see also Supplementary Movies 6 and 7). Data were representative of two similar independent experiments. **i** 2PM images of fibrillar co-assemblies in TAMRA-fIAPP-cross-seeded Aβ42/Nle3-VF mixtures containing HiLyte647-Aβ42/Fluos-Nle3-VF (50%) (1.5 h; incubations as in **f**); scale bars: 10 μm. Upper panel, fIAPP covered by Aβ42, Nle3-VF, and Aβ42/Nle3-VF (co-)assemblies and surrounded by amorphous or round/elliptical co-assemblies (see also **j** and Supplementary Movie 8). Lower panel, huge ternary nanofiber co-assembly (see also Supplementary Movie 9). Data were representative of two similar independent experiments. **j** 3D reconstruction of z-stacks/still images of fibrous co-assemblies shown in **i**/upper panel (see Supplementary Movie 8). The white arrow and dashed line in the image on the top indicate view of the section shown below; yellow arrows, round/elliptical co-assemblies; red arrow, fIAPP; blue and green arrows, Aβ42 & Nle3-VF bound to fIAPP; encircled area indicates Aβ42/Nle3-VF co-assembly bound to fIAPP. Scale bars, 10 μm (top), 1 μm (bottom). Data were representative of two similar independent experiments.

to control their formation/storage/disassembly related to their diverse biological functions[1,45,53,58–60]. Examples are amyloids from certain secreted peptide hormones or from proteins forming reversible subcellular condensates[45,53,58,60]. By contrast, most "pathological amyloids" are linked to cell damage and characterized by high stability and resistance to proteolysis[1,45,58]. Furthermore, increasing evidence suggests that amyloid fold polymorphism underlies amyloid pathogenicity and functional diversity[45,53,61]. Our results suggest that, in addition to pathogenic IAPP/Aβ hetero-amyloids generated by fibril-mediated cross-seeding, potentially beneficial IAPP/Aβ hetero-amyloids, such as those mimicked by IAPP/ACM nanofibers, might also exist and this could be also the case for other cross-interacting amyloids[1,16,62]. The structural characterization of IAPP/ACM nanofibers should help in identifying molecular factors that may redirect cytotoxic amyloid self-assembly into non-toxic and labile hetero-amyloids and enable the exploitation of amyloid fold versatility to design effective anti-amyloid molecules.

In conclusion, our work offers a series of designed peptides as highly potent inhibitors of amyloid self-assembly and reciprocal cross-seeding of IAPP and Aβ42 and as promising leads for effective anti-amyloid drugs in both T2D and AD. In addition, the identified nanofiber co-assemblies should guide the design of future functional (hetero-)amyloid-based supramolecular nanomaterials for biomedical and biotechnological applications[1,63].

## Methods

### Peptides and peptide synthesis

IAPP, IAPP-GI, rat IAPP, and their Nα-terminal fluorescein- or biotin-labeled analogs were synthesized by Fmoc-based solid phase synthesis (SPPS), subjected to air-oxidation, and purified by RP-HPLC as previously described in refs. 25, 26, 32, 64. Their stock solutions were prepared in 1,1,3,3,3,3-hexafluoro-2-isopropanol (HFIP) (4 °C), filtered over 0.2 μm filters (Millipore), and concentrations were determined by UV spectroscopy[32,44]. TAMRA-IAPP was synthesized by overnight coupling of 5,6-carboxytetramethylrhodamine (TAMRA) (Novabiochem/Merck) to RINK-resin-bound IAPP using a threefold molar excess of 2-(1H-Benzotriazol-1-yl)-1,1,3,3-tetramethyluronium-hexafluorophosphate (HBTU) and a 4.5 molar excess of $N,N$-diisopropylethylamine (DIEA) in $N,N$-dimethylformamide (DMF). TAMRA-IAPP cleavage from the resin and RP-HPLC purification were performed as for the other labeled IAPP analogs; stocks were made in HFIP (4 °C). Aβ42 was synthesized on Tentagel R PHB resin (0.18 mmol/g; Rapp Polymere) by Fmoc-SPPS using previously reported protocols[26,32,64]. Seed-free aqueous Aβ42 stock solutions (10–20 μM) were obtained by SEC performed according to published protocols[65,66]. Briefly, HPLC-purified Aβ42 (purification by Peptide Specialty Laboratories) was dissolved (1 mg/ml) in a solution of 5 M GdnHCl in 10 mM TRIS/HCl pH 6.0 and loaded onto a Superdex 75 10/300 GL column (eluent: 50 mM ammonium acetate pH 8.5, 0.5 ml/min). The monomeric Aβ42 elution peak was collected on ice, stored at 4 °C, and used within 1 week; peptide concentration was determined by UV spectroscopy. Fluorescein-isothiocyanate-β-Ala-labeled Aβ42 (FITC-Aβ42) and TAMRA-labeled Aβ42 (TAMRA-Aβ42) were from Bachem and HiLyte647-Aβ42 from AnaSpec; their stocks were prepared in HFIP (4 °C).

All Aβ(15–40) analogs comprising ACMs, non-inhibitors, and partial segments thereof (Supplementary Tables 1, 4) were synthesized using previously described standard Fmoc-SPPS protocols[26,32,64] and, in most cases, WANG-resin (0.3–0.5 mmol/g; Iris Biotech); Tentagel R PHB resin was used for Nle3, R3, and G3-VF (0.16 mmol/g; Rapp Polymere). Briefly, double couplings were usually performed using three-fold molar excess protected amino acid and HBTU and 4.5-fold molar excess of DIEA in DMF. For difficult couplings, we applied either 2-(7-aza-1H-benzotriazole-1-yl)-1,1,3,3-tetramethyluronium-hexafluorophosphate (HATU) or four to sixfold molar excess of protected amino acids, and/or triple couplings. N-terminal fluorescein-labeled Aβ(15–40) analogs were synthesized by coupling peptide-resins with 5,6-carboxyfluorescein (Sigma-Aldrich) using threefold molar excess protected amino acid and HATU and 4.5-fold molar excess of DIEA (double couplings). N-terminal Atto647N-labeled Nle3-VF was synthesized by coupling peptide-resin with Atto647N (carboxy-derivative) (ATTO-TEC) using HATU. Peptide cleavage from the resin was performed with 95% TFA/H₂O. All peptides were purified by RP-HPLC on Nucleosil 100 C18 (Grace) or Reprosil Gold 200 C18 columns (Dr. Maisch) according to previously described protocols[64,67]. Stock

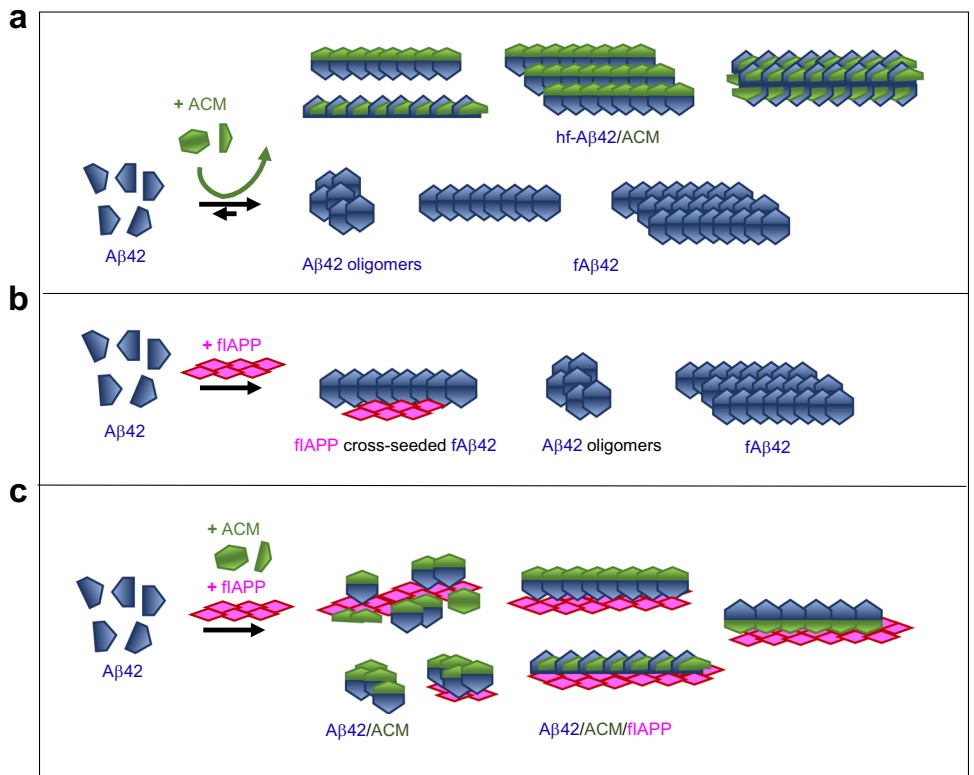

**Fig. 8 |** Schematic overview of identified co-assemblies and proposed mechanisms of ACM-mediated suppression of Aβ42 amyloid self-assembly (**a**) and its cross-seeding by fIAPP (**b, c**). **a** Lower row, Aβ42 self-assembles into toxic oligomers and fAβ42. Upper row, non-toxic ACMs bind with low nanomolar affinity Aβ42 and redirect it into heteromeric nanofiber co-assemblies (hf-Aβ42/ACM), which are non-toxic, seeding-incompetent, and thermolabile and become easier degraded and more effectively phagocytosed than fAβ42. hf-Aβ42/ACM, which may form by lateral or axial co-assembly are shown. **b** Cross-seeding of Aβ42 by fIAPP yields via secondary nucleation fAβ42/fIAPP co-assemblies, fAβ42, and toxic Aβ42 oligomers. **c** ACM-mediated inhibition of cross-seeding of Aβ42 by fIAPP. Non-toxic ACMs and ACM/Aβ42 co-assemblies (both fibrillar and amorph) bind to fIAPP yielding non-toxic and cross-seeding-incompetent fibrous co-assemblies.

solutions were made in HFIP (4 °C); peptide concentrations were determined by peptide weight or by UV spectroscopy (fluorescently-labeled analogs)[25,32].

All synthetic peptides were characterized by matrix-assisted laser desorption ionization (MALDI-MS) or electrospray ionization (ESI-MS) mass spectrometry (Supplementary Table 4).

### Thioflavin T (ThT) binding assays

**IAPP fibrillogenesis-related studies.** Effects of the different peptides on IAPP fibrillogenesis, including self- and cross-seeded fibrillogenesis, were studied in combination with TEM and MTT reduction assays according to previously established ThT binding assay systems[12,20,21,32]. At the indicated time points, aliquots of peptide incubations (made as described below) were gently mixed with the ThT solution (20 µM ThT in 0.05 M glycine/NaOH, pH 8.5, if not stated otherwise) in a 96-well black MTP (FluoroNunc/Thermo Fisher Scientific). ThT binding was determined immediately by measuring fluorescence emission at 486 nm following excitation at 450 nm using a 2030 Multilabel Reader VictorX3 instrument (PerkinElmer Life Sciences) (software: PerkinElmer 2030 Manager (V4.0))[20]. ThT binding of seeds/buffer were subtracted from the data in all assays related to seeding events; in all other cases, raw or normalized data were shown if not stated otherwise. All IAPP-related incubations were performed at 20 °C except for the (cross-)seeded ones (RT). Preformed fIAPP were generally prepared by incubating IAPP (12 or 16.5 µM) in ThT buffer for 3–9 days (20 °C), quantified by ThT binding, and verified by TEM[12,20].

Peptide incubations were performed as follows: For studying effects on IAPP fibrillogenesis, freshly made IAPP (16.5 µM) and IAPP/peptide mixtures were incubated in 50 mM sodium phosphate buffer, pH 7.4, with 100 mM NaCl containing 0.5% HFIP (abbreviated "ThT

buffer") for up to 7 days[12,32]. For studying effects on fIAPP-mediated seeding of IAPP fibrillogenesis, preformed fIAPP (10%) were added to freshly made IAPP (12 µM) or IAPP/peptide mixtures (1/2) in ThT buffer; solutions were incubated for several days as indicated. To determine the detection limit of the IAPP-related ThT binding assay, fIAPP were first made by incubating IAPP (16.5 µM) in ThT buffer (7 days). Following fIAPP quantification by ThT binding and verification by TEM (Figs. 1d, 2f), serial fIAPP dilutions were made and ThT binding was measured as described in the first paragraph. The significance of differences between ThT binding of the various fIAPP amounts versus buffer alone was analyzed by one-way ANOVA and Bonferroni's Multiple Comparison test. To investigate whether ACM binding to fIAPP surfaces might compete with ThT binding, ThT binding of preformed fIAPP (16.5 µM, ThT buffer, 4 day-aged) and hf-IAPP/Nle3-VF (16.5 µM, ThT buffer, 4 day-aged) was determined using ThT solutions containing 20 or 200 µM ThT. To investigate whether the binding of ACMs to fIAPP surfaces ("ACM coating" of fIAPP) might block ThT binding of fIAPP, ACM (twofold) was added to preformed fIAPP (16.5 µM); ThT binding of ACM/fIAPP mixtures and non-treated fIAPP was determined before and following co-incubation of ACM with fIAPP (1 day) as above. To investigate the effects of Nle3-VF on already nucleated IAPP fibrillogenesis, aliquots of an IAPP incubation (16.5 µM in ThT buffer) were added to the ACM at different time points of fibrillogenesis as described[12,21]. To investigate whether hf-IAPP/ACM co-assemblies might seed IAPP fibrillogenesis, seed amounts (10%) of hf-IAPP/ACM (made by incubating IAPP (16.5 µM) with ACM (twofold) for 7 days in ThT buffer) were added to freshly made IAPP (12 µM in ThT buffer); incubations were performed for 48 h. Seeding effects of preformed fIAPP (10%) made under the same conditions as hf-IAPP/ACM were studied in parallel. Dot blot analysis (Supplementary Fig. 9a) confirmed

that similar amounts of fIAPP and hf-IAPP/ACM were used for seeding or other assays. To determine the effects of ACMs on fAβ42-mediated cross-seeding of IAPP fibrillogenesis, seed amounts (10%) of fAβ42 (made by incubating Aβ42 (88 μM) in ThT buffer containing 1% HFIP for 19 days at 37 °C; fibril formation confirmed by ThT binding and TEM) were added to freshly made IAPP (12 μM in ThT buffer) or IAPP/peptide mixtures (1/2); incubations were performed for 48 h.

**ACM fibrillogenesis-related studies.** To study the fibrillogenic potential of ACMs, peptides, and Aβ40 (positive control) (100 μM) were incubated in 10 mM aqueous sodium phosphate buffer, pH 7.4 (1% HFIP) for 4 days[21,32]. ThT fluorescence was measured at 0 h and 4 days by mixing an aliquot with a ThT-containing solution (121 μM ThT, 0.05 M glycine/NaOH, pH 8.5); buffer values were subtracted from the data shown in Supplementary Fig. 5b. The absence of fibrils and cytotoxic aggregates from these solutions was confirmed by TEM (Supplementary Fig. 5a) and MTT reduction assays (Supplementary Fig. 5c).

**Aβ42 fibrillogenesis-related studies.** To study the effects of the different peptides on Aβ42 fibrillogenesis, synthetic Aβ42 isolated from SEC (see "Peptides and peptide synthesis") was used. Peptide incubations were performed in the presence of ThT in 96-well black MTPs (FluoroNunc, Thermo Fisher Scientific) based on previously developed protocols[65]. Incubation conditions for all assays were (if not stated otherwise): Aβ42 (5 μM) alone or its mixture with the peptide (at the indicated molar ratios) in 45 mM ammonium acetate, pH 8.5, containing 10 μM ThT (37 °C); MTPs were shaken (500 rpm; orbital shaker (CAT S20)) for the first 5 h of the fibrillogenesis. ThT fluorescence was measured with a 2030 Multilabel Reader VictorX3 instrument at the indicated time points as under IAPP-related assays. Values of seeds or buffer alone were subtracted from the data in self-/cross-seeding assays; all other data shown are raw data except for data in Supplementary Figs. 25b, 27a, which were normalized).

For studying the effects of ACMs on fAβ42-mediated seeding of Aβ42, preformed fAβ42 (made by incubating Aβ42 (5 μM) as above but without ThT for 6 days (TEM, Fig. 6d)) were added to freshly made Aβ42 (5 μM) or Aβ42/ACM mixtures (5 μM each; made on ice) just before the addition of ThT (10 μM). fAβ42 seed concentration was 0.5 μM (10%) and incubations (37 °C) were performed as above. Of note, fAβ42 were quantified/verified by ThT binding, dot blots, and TEM. For studying whether hf-Aβ42/ACM might seed Aβ42 fibrillogenesis, Aβ42 (5 μM) and Aβ42/ACM mixtures (5 μM each) were first incubated for 6 days without ThT as described in the top section to obtain fAβ42 and hf-Aβ42/ACM; fibrils were quantified/verified by ThT binding (fAβ42), TEM (fAβ42 and hf-Aβ42/ACM), and dot blots. Seed amounts of fAβ42 or hf-Aβ42/ACM (10%, 0.5 μM) were then mixed (on ice) with freshly made Aβ42 (5 μM) in 45 mM ammonium acetate, pH 8.5, and following the addition of ThT (10 μM) incubations (37 °C) were performed as described above. To study the effects of ACMs added at post-nucleation time points of Aβ42 fibrillogenesis (including effects on preformed Aβ42 oligomers (2 h-aged Aβ42)), Aβ42 (5 μM) was incubated in the presence of ThT as described in the top section and mixed with the ACM (1/1) at the indicated time points of fibrillogenesis. Effects of ACMs on fIAPP-mediated cross-seeding of Aβ42 were studied as follows: Aβ42 (10 μM) alone and Aβ42/ACM mixtures (1/2) were prepared as above on ice. Preformed fIAPP (2 μM) (made by incubating IAPP (128 μM) in ThT buffer for 9–12 days; fIAPP quantified/verified by ThT and TEM) were added to the above solutions just prior to the addition of ThT (10 μM). The final composition of the assay buffer was: 45 mM ammonium acetate, pH 8.5, containing 10 μM ThT and <2% of ThT buffer resulting from the fIAPP seed or the buffer alone solution (in the Aβ42 without seed (control) solution). Incubations (37 °C) and determination of ThT fluorescence were performed as for all other Aβ42-related studies.

**Assessment of cell damage via the MTT reduction assay**
Studies on the effects of peptides on the formation of cell-damaging IAPP assemblies were performed in cultured RIN5fm (obtained from T.E. Rucinsky at the Washington University Tissue Culture Support Center) in combination with the ThT binding assay and TEM using previously established protocols[12,20,21,32,68]. Briefly, cells were cultivated and platted in 96-well plates as described in ref. 68. Aliquots of solutions used for the ThT binding assays (see "ThT binding assays") were diluted with cell medium at the indicated incubation time points (24 h or 7 days) and added to the cells at the indicated final concentrations. Following incubation with the cells for ~20 h (37 °C, humidified atmosphere, 5% CO$_2$), cell damage was assessed by measuring cellular MTT reduction as described[32,68]. IC$_{50}$ values were determined as described[20,21,32]. Briefly, IAPP (16.5 μM) was incubated with different molar ratios of the ACMs in ThT buffer (see under ThT binding assay) for 24 h and solutions were added to the cells (IAPP, 100 nM); cell viability was assessed as above. To determine cell-damaging effects of ACM-coated fIAPP, preformed fIAPP (16.5 μM) was co-incubated with the ACM (33 μM) for 1 day as under ThT-binding assays. Solutions of ACM-coated fIAPP versus fIAPP alone were diluted with cell medium and incubated with the cells (fIAPP, 500 nM) as described above.

The studies on the effects of ACMs on the formation of cell-damaging Aβ42 assemblies were performed in combination with the ThT binding assay and TEM using PC12 cells obtained from DSMZ (German Collection of Microorganisms and Cell Cultures) (DSMZ no. ACC 159) and cultured and plated as described[12]. Incubations of Aβ42 alone and its mixtures were made in MTPs as described for the ThT binding assays (parallel to the incubations made for the ThT binding assay) but without ThT; 6-day-aged solutions (37 °C) were diluted with cell medium and added to the PC12 cells at the indicated final concentrations. Following incubation with the cells for ~20 h (37 °C, humidified atmosphere, 5% CO$_2$), cell damage was assessed by measuring cellular MTT reduction as described[12]. To determine IC$_{50}$ values of the effects of ACMs, incubations of Aβ42 (5 μM) or its mixtures with various amounts of the ACMs were performed for the ThT binding assay (37 °C) but without ThT in MTPs. 6-day-aged incubations were diluted with medium and added to the PC12 cells (Aβ42, 1 μM), and cell damage was assessed following 20 h incubation with the cells as above. On note, anomalous concentration-dependence profiles were found for mixtures of Aβ42 with L3-LF and Nle3-LF, most likely due to aggregation; therefore, IC$_{50}$ values were not determined.

To study the effects of Nle3-VF on cytotoxicity of preformed (TAMRA-)Aβ42 oligomers, we first studied cytotoxicity of (TAMRA-) Aβ42 solutions (5 μM) at different early time points of fibrillogenesis by the MTT reduction assay (Supplementary Fig. 27b, d). Aliquots of the incubations (37 °C, shaking 500 rpm; prepared as for the ThT binding assay but without ThT) were diluted with medium (Aβ42 1 μM), added to PC12 cells at 0, 2, 4, 6, and 24 h, and following incubation with the cells for 20 h cell damage was assessed as described above. For the incubations, Aβ42 isolated from SEC was used as for all other assays while TAMRA-Aβ42 was used as a dry film from its HFIP stock. Based on the results of the ThT binding assay of Aβ42 (Supplementary Fig. 27a), the MTT reduction assay of (TAMRA-)Aβ42 (Supplementary Fig. 27b, d), dot blot analysis of both (Supplementary Fig. 27c) and TEM, 2 h-aged (TAMRA-)Aβ42 (5 μM) (abbr. (TAMRA-) oAβ42) was found to consist mostly of cytotoxic oligomers. For studying effects of the addition of Nle3-VF to preformed (TAMRA-) oAβ42, Nle3-VF was mixed with 2 h-aged (TAMRA-)Aβ42 (1/1) (solutions made as for the ThT binding assay but without ThT). Preformed (TAMRA-)oAβ42 and their mixtures with Nle3-VF were then incubated for an additional 2 h (37 °C, shaking 500 rpm; conditions as for the ThT binding assay). Incubation with the cells and the MTT reduction assay were performed as above. Statistical analysis (Supplementary

Fig. 27d) was performed by one-way ANOVA and Bonferroni (software: OriginPro 2021).

To study the effects of ACMs on fIAPP-mediating cross-seeding of formation of cell-damaging Aβ42 assemblies, incubations were made for the corresponding ThT binding assays but without ThT. Solutions were aged for 1.5 h (37 °C, shaking at 500 rpm). Incubation with the cells and the MTT reduction assay were performed as described above.

Effects of ACMs on PC12 cell viability were studied using the 4 day-aged solutions applied in the ThT binding assays, which were performed to determine their amyloidogenic potential (see under "ThT binding assays"). Following incubation with the cells (at 20 μM) for ~20 h, cell damage was assessed by MTT reduction; data were corrected for buffer effects. For comparison, effects of aged Aβ40 were also studied and cytotoxicity was as expected[12].

### Transmission electron microscopy (TEM)
Aliquots of solutions used for ThT binding, MTT reduction, or other assays were applied on formvar/carbon-coated grids at the indicated incubation time points. Grids were washed with ddH₂O and stained using aqueous 2% (w/v) uranyl acetate as described in ref. 44. Examination of the grids was done with a JEOL 1400 Plus electron microscope (120 kV) (data acquisition software: TEM Center (v. 1.7.19.2439; data analysis with Image J (1.50i)). For Aβ42-related studies, solutions made as for the ThT binding assay but without containing ThT were used for TEM and the MTT reduction assays. Kinetics of evolution of IAPP homo- and IAPP/Nle3-VF hetero-fibrils from amorphous aggregates was followed by TEM in solutions made in 10 mM sodium phosphate buffer, pH 7.4 (Fig. 4a) and also in solutions made in ThT buffer and very similar results were found.

### Immunogold-TEM
Immunogold-TEM was performed based on previously described protocols[13]. Briefly, peptide solutions made as described for the corresponding ThT binding assays were applied onto the grids at the indicated incubation time points. Grids were blocked with 0.1% BSA in 1xPBS. fIAPP was detected with a fibril-specific mouse anti-fIAPP antibody (Synaptic Systems; Cl. 91E7)[23]. Nle3-VF was revealed by a rabbit anti-Aβ40 polyclonal antibody (Sigma-Aldrich) exhibiting 10–20% NSB to IAPP. The two antibodies (in 0.1% BSA in 1xPBS; dilution 1/10) were deposited simultaneously onto the grid and incubated for 20 min. Following washing with 1xPBS, grids were incubated (20 min) with secondary antibodies goat anti-rabbit gold-conjugate (10 nm) and goat anti-mouse gold-conjugate (5 nm) (Sigma-Aldrich) (in 0.1% BSA in 1xPBS, dilution 1/10) as above. Following 1xPBS and ddH₂O washings, uranyl acetate staining and grid examination were performed as described under "TEM". To quantify IAPP and Nle3-VF contents of fibrils, 5 and 10 nm gold particles were counted (software: Image J (V1.50i)); "antibody reactivity" is expressed as % of a total number of gold particles bound.

### Far-UV CD spectroscopy
CD spectra were recorded using a Jasco 715 spectropolarimeter (software: Spectra Manager V1.55.00 (Build 2)). CD spectra (average of three spectra) were measured between 195–250 nm, at 0.1 nm intervals, with a response time of 1 s, and at RT. The spectrum of the buffer was always subtracted from the spectra of the peptide solutions. Peptide incubations related to ACM alone or ACM/IAPP interactions were performed according to previously established protocols[12,20,32]. Briefly, to study peptide conformations and oligomerization propensities, CD spectra of freshly made solutions in 10 mM sodium phosphate buffer, pH 7.4, containing 1% HFIP were measured at 5 μM or at the indicated concentrations in concentration-dependence studies. For studying hf-IAPP/ACM, IAPP (16.5 μM) was incubated with Nle3-VF or VGS-VF (33 μM) in ThT buffer

for the ThT binding assay for 7 days and spectra were measured at the indicated time points. For comparison, spectra of IAPP, Nle3-VF, and VGS-VF alone were also measured. For studying the structure of hf-Aβ42/Nle3-VF, incubations were performed for the ThT binding assays but in the absence of ThT. Briefly, Aβ42 alone (5 μM), Nle3-VF alone (5 μM), and their mixtures (5 μM each) in 45 mM ammonium acetate (pH 8.5) were incubated for 6 days at 37 °C and CD spectra were measured.

### Fluorescence spectroscopic titration assays
Fluorescence spectroscopic studies were performed with a Jasco FP-6500 fluorescence spectrophotometer (software: Spectra Manager V1.54.03 (Build 1)) using previously established protocols[12,20,25,26,32]. Briefly, excitation was at 492 nm and spectra were measured between 500 and 600 nm. All titrations were performed in freshly made solutions of synthetic N-terminal fluorescently-labeled peptide (5 nM) and various amounts of unlabeled peptide in 10 mM sodium phosphate buffer, pH 7.4 (1% HFIP) within 2–5 min following solution preparation as described in refs. 20, 25, 26. Under these experimental conditions, freshly made solutions of Fluos-IAPP and FITC-Aβ42 (5 nM) consist mostly of monomers[25,26,32] and the same was found for Fluos-ACMs (5 nM) (e.g., Supplementary Fig. 5e). Apparent binding affinities (app. $K_D$s) were estimated by using 1/1 binding models (software: OriginPro 2016G, Origin 2021, and GraFit (v.5)) as described in refs. 20, 25, 26, 32. However, due to the high self-assembly propensities of involved peptides, more complex models might also apply. Determined app. $K_D$s are means (±SD) from three binding curves derived from three independent titration assays.

### Cross-linking, NuPAGE, and Western blot (WB)
Hetero-complex cross-linking studies in combination with NuPAGE and WB were performed with a previously developed assay system used for the characterization of Aβ and IAPP homo- and hetero-assemblies[12,20,21]. Briefly, for characterizing IAPP homo-/hetero-assemblies, IAPP (30 μM), IAPP/ACM mixtures (1/2), and ACMs alone (60 μM) were incubated in 10 mM sodium phosphate buffer (pH 7.4) for up to 7 days (20 °C). In the case of Aβ42 homo-/hetero-assemblies, Aβ42 (30 μM) and Aβ42/ACM mixtures (1/2) were incubated in 10 mM sodium phosphate buffer (pH 7.4) for up to 6 days. At the indicated time points (0 h, 24 h, and 7 days (IAPP studies) or 0 h, 3 h, 24 h, and 6 days (Aβ42 studies)), aliquots were cross-linked (2 min) with 25% aqueous glutaraldehyde (Sigma-Aldrich) and treated with a 2 M NaBH₄ solution (in 0.1 M NaOH, 20 min). Following precipitation with trichloroacetic acid (10%) (4 °C) and centrifugation (10 min, 12000 g), pellets were dissolved in reducing NuPAGE sample buffer, boiled (5 min, 95 °C) and subjected to NuPAGE gel electrophoresis as described using 4–12% Bis-Tris gels and MES running buffer (Thermo Fisher Scientific)[12,20]. Equal amounts of IAPP or Aβ42 were loaded in all lanes. Peptides were transferred onto nitrocellulose membranes (XCell II Blot Module, Thermo Fisher Scientific). Membranes were blocked overnight (10 °C) with 5% milk in TBS-T (20 mM Tris/HCl, 150 mM NaCl, and 0.05% Tween-20). To reveal homo-/hetero-assemblies, membranes were incubated (2 h) with one of the following primary antibodies (in 5% milk in TBS-T): rabbit polyclonal anti-IAPP (Peninsula; 1:1000) for IAPP-containing assemblies, rabbit polyclonal anti-Aβ40 (Sigma-Aldrich; 1:2000) for ACM-containing assemblies, or mouse monoclonal anti-Aβ(1–17) (6E10, BIOZOL; 1:2000) for Aβ42-containing assemblies (no cross-reactivity with ACMs). Primary antibodies were combined with suitable peroxidase (POD)-coupled secondary antibodies (donkey anti-rabbit-POD (1:5000) or goat anti-mouse-POD (1:1000)) and homo-/hetero-assemblies were revealed with Super-Signal West Dura Extended Duration Substrate (Thermo Fisher Scientific) (visualization by LAS-4000 mini (Fujifilm); software: Image Reader LAS-4000 mini (V2.0)). Membranes were stripped by incubating in stripping buffer (2% SDS, 100 mM β-mercaptoethanol, 50 mM

TRIS, pH 6.8) for 20 min at 60 °C and at RT for 45 min. Prestained protein size markers ranging from 3.5 to 260 kDa (Invitrogen) were electrophoresed in the same gels.

## Size exclusion chromatography (SEC)

SEC was performed with a Superdex 75 10/300 GL column (GE Healthcare) using an Ultimate 3000 HPLC device (Thermo Scientific) (software: Chromeleon (v7)). The flow rate was 0.5 ml/min and detection was at 214 nm. For IAPP-related studies, elution buffer was 50 mM sodium phosphate buffer, pH 7.4, containing 100 mM NaCl. IAPP (16.5 µM) or IAPP/ACM (or IAPP/VGS-VF) mixtures (1/2) were incubated in ThT buffer as for the ThT binding assays and at the indicated incubation time points centrifuged (1 min, 20,000 × g) and loaded onto the column. For SEC of synthetic Aβ42 to produce seed-free aqueous Aβ42 solutions, see under "Peptides and peptide synthesis". For Aβ42-related studies, elution buffer was 50 mM ammonium acetate, pH 8.5. Aβ42 (5 µM) or Aβ42/ACM (1/1) mixtures were incubated under ThT binding assay conditions (without ThT) and loaded onto the column at indicated time points. The column was calibrated with proteins/peptides of known molecular weights.

## ESI-IMS-MS

A Synapt XS HDMS mass spectrometer (Waters) equipped with an Acquity Premier nano-ESI interface (Waters) was used (software: MassLynx (v4.2)). MS spectra were obtained using the following parameters: Positive mode nano-ESI; capillary voltage of 1.7 kV; nitrogen nebulizing gas pressure 0.8 psi; cone voltage 40 V; source temperature 60 °C; backing pressure 1.6 mBar; Trap Collision Energy Control, ramping 10–100 V; ramped traveling wave height 7–20 V; IMS nitrogen gas flow 20 ml/min, traveling wave speed 300 m/s and IMS cell pressure 0.55 mBar. The m/z scale was calibrated with NaJ cluster ions and Glufib (m = 785.8427) as lock mass. For analysis, freshly made solutions of IAPP (30 µM), Nle3-VF (60 µM), and IAPP/Nle3-VF (1/2) in 10 mM sodium phosphate buffer (pH 7.4) were used. The samples were vortexed and directly applied to nano-ESI, operating the nano-HPLC-system at a constant flow rate of 0.2 ml/min in bypass mode for direct sample injection. MS analysis was carried out ~2 min after sample preparation.

## ANS binding fluorescence spectroscopy

ANS binding studies were performed with a Jasco FP-6500 fluorescence spectrophotometer (software: Spectra Manager V1.54.03 (Build 1)) as previously described in refs. 20, 44. Briefly, excitation was at 355 nm and fluorescence emission spectra were recorded between 355 and 650 nm. Solutions of ANS alone (8 µM) and its mixtures with IAPP (2 µM) or IAPP/Nle3-VF (1/2) mixtures were freshly made in 10 mM sodium phosphate buffer, pH 7.4, containing 1% HFIP, and spectra were recorded at the indicated time points.

## Congo red (CR) binding

Assays were performed based on previously described protocols[44,68,69]. Solutions of IAPP (16.5 µM), Aβ42 (5 µM), and their mixtures with ACMs (IAPP/ACM, 1/2; Aβ42/ACM, 1/1) were prepared as described for the ThT binding assay (Aβ42 without ThT). After 7-day- (IAPP-related samples) or 6-day-aging (Aβ42-related samples), solutions consisted mostly of fibrillar assemblies (Figs. 1, 2) and were used for studying CR binding. For the CR spectral shift assay[69], 7-day-aged IAPP or IAPP/ACM or buffer alone solutions were mixed with CR (7 µM) and CR absorption spectra between 400–700 nm were measured using a Clariostar Plus MTP reader (BMG Labtech) (software: Clariostar Software (v. 5.70 R3)). Spectra are presented after subtraction of a spectrum taken in the absence of CR, to account for light scattering effects. For CR staining studies, solutions were spotted on a glass slide, air-dried, stained with 200 µM CR in 80% EtOH, and examined between a cross-polarizer/analyzer with an Olympus CKX41 light microscope[44].

## Hetero-complex pull-down assays

Pull-down assays were performed using streptavidin-coupled magnetic beads (Dynabeads M-280 Streptavidin, Dynal) as described in refs. 12, 32. Briefly, solutions of Biotin-IAPP (16.5 µM), Biotin-IAPP/Nle3-VF-mixtures (1/2), and Nle3-VF (33 µM; control for non-specific binding (NSB) to beads) in 10 mM sodium phosphate buffer, pH 7.4 were aged for 7 days (20 °C) and subsequently incubated with the beads for 4 h at RT. Bead-bound complexes were isolated by magnetic affinity. Following washing, beads were boiled with reducing NuPAGE sample buffer (5 min, 95 °C) and supernatants subjected to NuPAGE electrophoresis and WB as described under "Cross-linking, NuPAGE and WB". Equal amounts were loaded in all lanes; lane "Nle3-VF (control)", freshly dissolved Nle3-VF without incubation with the beads. Scans of uncropped blots are provided in the Source Data file.

## CLSM and STED imaging

IAPP or IAPP/ACM mixtures (1/2) containing 10% N-terminal fluorescently-labeled analogs TAMRA-IAPP and Atto647N-ACM (IAPP(total), 16.5 µM; ACM(total), 33 µM) were incubated in 10 mM sodium phosphate buffer, pH 7.4 for 7 days (20 °C). Aliquots (30–40 µl) were pipetted onto SuperFrost Plus adhesion slides (Thermo Fisher Scientific), air-dried, covered with a high precision coverslip (#1.5; Ibidi), and embedded using Prolong Diamond Antifade Mountant (Thermo Fisher Scientific). CLSM and STED were performed using a Leica SP8 STED 3X microscope (HC PL APO 93x/1.30 GLYC CORR STED objective) with a tunable white light laser source to excite fluorophores[70]. Depletion power (660 nm (TAMRA), 775 nm (Atto647N)), and time-gated detection of excited light were chosen to minimize sample damage while optimizing xyz-resolutions. Images were collected in a sequential scanning mode (hybrid-diode detectors) to maximize signal collection while minimizing channel cross-talk (TAMRA: excitation 552 nm/emission 557–645 nm; Atto647N: excitation 646 nm/emission 651–700 nm). 3D reconstructions/fibril measurements were performed using Leica's LAS-X software package (v1.2). Datasets were deconvoluted using Leica's LIGHTNING application.

## 2PM and FLIM-FRET studies

Solutions analyzed by 2PM or FLIM-FRET consisted of either N-terminal fluorescently-labeled peptides (100%) or mixtures of labeled with non-labeled peptides (when indicated) and were prepared as follows: For most IAPP-related studies, hf-IAPP/ACM were prepared by incubating TAMRA-IAPP (16.5 µM) with synthetic N-terminal fluorescein- or Atto647N-labeled ACMs (33 µM) (as indicated in the figure/figure legend) in 10 mM sodium phosphate buffer, pH 7.4 (abbreviated "1xb") for 6–7 days (20 °C). For comparison, aged TAMRA-IAPP (16.5 µM) was also examined and consisted mostly of fibrillar assemblies (TAMRA-fIAPP; Supplementary Fig. 12d). In some cases (i.e. samples examined by both STED and 2PM), IAPP/ACM mixtures (6–7 days aged, 1xb (20 °C)) consisting of 16.5 µM IAPP(total) and 33 µM ACM(total) with each of them containing 10% of labeled peptide were used as indicated. For 2PM studies related to ACM-coated fIAPP (Supplementary Fig. 13e), first, fIAPP were made by incubating TAMRA-IAPP (16.5 µM) in 1xb for 48 h. fIAPP were then co-incubated with a mixture of Nle3-VF/Atto647-Nle3-VF (9/1; 33 µM Nle3-VF(total)) in 1xb for 1 day (20 °C) to yield ACM-coated fIAPP. For 2PM and FLIM-FRET studies regarding the role of monomeric/prefibrillar IAPP on the formation of IAPP/Nle3-VF nanofiber co-assemblies (Fig. 4c, d and Supplementary Fig. 16), aliquots from a freshly made mixture of Fluos-Nle3-VF (33 µM) with TAMRA-IAPP (1.65 µM) in 1xb (20 °C) were examined at indicated incubation times. For the corresponding studies on the role of fIAPP (Supplementary Fig. 17), preformed TAMRA-fIAPP seeds (made by incubating TAMRA-IAPP (16.5 µM) in 1xb for 5 days (20 °C)) were added to Fluos-Nle3-VF (33 µM); the final concentration of TAMRA-fIAPP was 3.3 µM. For FLIM-FRET studies of the different TAMRA-IAPP/Fluos-Nle3-VF co-assemblies (Figs. 3j, 4d and

Supplementary Fig. 17a–c), the donor Fluos-Nle3-VF alone (data shown in Supplementary Figs. 14, 16, 17d, e) was incubated under the same experimental conditions as the corresponding co-assemblies.

For all Aβ42-related studies, solutions consisted of 1/1 mixtures of unlabeled/labeled peptides (as indicated) and were prepared as for the ThT binding assays (without ThT) (45 mM ammonium acetate, pH 8.5, 37 °C; shaking between 0–5 h; aging as indicated) as follows: For the studies on fAβ42 versus hf-Aβ42/ACM, Aβ42 solutions (Aβ42(total) 5 μM) consisted of 50% TAMRA-Aβ42 and 50% Aβ42 (6-day-aging); Aβ42/ACM mixtures (1/2) contained, in addition to Aβ42/TAMRA-Aβ42 (1/1; Aβ42(total), 5 μM), ACM/Fluos-ACM (1/1; ACM(total), 10 μM) (4–6-day-aging). For FLIM-FRET analysis of hf-TAMRA-Aβ42/Fluos-Nle3-VF (Fig. 6f and Supplementary Fig. 23), Fluos-Nle3-VF alone (mixture of Fluos-Nle3-VF/Nle3-VF (1/1), Nle3-VF(total) 10 μM) was incubated under the same experimental conditions (see above; 4 day-aging) as the TAMRA-Aβ42/Fluos-Nle3-VF (1/2) mixture (consisting of TAMRA-Aβ42/Aβ42, (Aβ42(total) 5 μM) and Fluos-Nle3-VF/Nle3-VF (1/1) (Nle3-VF(total), 10 μM). Of note, no time-dependent pH changes were observed in buffer solutions incubated under the experimental conditions applied for peptide incubations used for FLIM-FRET studies. For studies on fIAPP-mediated cross-seeding of Aβ42, the Aβ42 alone solution (Aβ42(total), 10 μM) contained HiLyte647-Aβ42 (50%); the Aβ42/ACM (1/2) mixtures consisted of Aβ42/HiLyte647-Aβ42 (1/1) (Aβ42(total), 10 μM) and Nle3-VF/Fluos-Nle3-VF (1/1) (Nle3-VF(total), 20 μM); solutions were aged for 1.5 h. TAMRA-fIAPP seeds for cross-seeding were prepared by incubating TAMRA-IAPP (128 μM) in ThT buffer for 6 days (20 °C)); their concentration in the cross-seeded solution was 2 μM.

Aliquots from the above mentioned samples were applied onto SuperFrost Plus adhesion slides, air-dried, washed (Aβ42-related studies), and embedded with Prolong Diamond Antifade Mountant as for CLSM and STED with one exception: hf-Aβ42/Nle3-VF co-assembly-related samples (incl. the sample of donor alone) (Fig. 6e, f and Supplementary Fig. 23), were centrifuged (20,000×g, 20 min) and pellets resuspended in 1xb just before application onto the slide to minimize interference of the sample preparation buffer. Samples were imaged with a two(multi)-photon Leica TCS SP8 DIVE multispectral two (multi)-photon microscope with 4TUNE NDD detection module, LIGHTNING adaptive deconvolution, and fast lifetime contrast (FALCON) modality, equipped with extended IR spectrum tunable laser (680–1300 nm) (New InSight® X3™, Spectra-Physics) and fixed IR laser (1045 nm), advanced Vario Beam Expander (VBE), ultra-high-speed resonance scanner (8 kHz), HC PL IRAPO 25x/1.0 WATER objective, and FLIM-FRET modality[70]. Images were collected in sequential scanning mode (hybrid-diode detectors; TAMRA: excitation 1100 nm/emission 560–630 nm; fluorescein (Fluos): excitation 920 nm/emission 480–550 nm; HiLyte647: excitation 1280 nm/emission 635–715 nm) and handled using Leica's LAS-X software package. Deconvolutions were performed using Huygens Professional or Leica's LIGHTNING application.

For fluorescence lifetime imaging (FLIM), up to 1000 photons/pixel were captured (time-correlated single-photon counting (TCSPC) mode). Samples were prepared as described above. Fluorescence decays were fit using Leica's FALCON software applying multi-exponential models. The quality of fits was assessed by randomly distributed residuals/low Chi-square values. The number of components (n) used for fittings was manually fixed to values ($n = 2–4$) that minimized Chi-square statistic. In control experiments, the fluorescence lifetime of the donor (Fluos-Nle3-VF) (prepared as the corresponding mixtures) in absence of the acceptor was acquired similarly (a multiexponential model was applied). Since both donor alone and its mixtures with the acceptor showed multiexponential decays, "amplitude-weighted average lifetime" (τAvAmp) was used to calculate FLIM-FRET efficiency. This lifetime value was calculated by the Leica FALCON software after fitting and specified according to the following

formula: $\tau AvAmp = \frac{\sum Ai^*\tau i}{\sum Ai}$ (with τAvAmp: amplitude-weighted average lifetime, A: amplitude, and τ: lifetime). By comparing the amplitude-weighted average lifetimes of the unquenched donor with the donor undergoing FRET, the software extracted the FLIM-FRET efficiency according to the following formula: $FRET\ EFF(E) = 1 - \frac{\tau AvAmp}{\tau D}$ (with τAvAmp: amplitude-weighted average lifetime of the quenched donor (undergoing FRET); τD, amplitude-weighted average lifetime of the unquenched donor).

## Dot blot assays to assess binding of ACMs to IAPP and Aβ42 fibrils and monomers

IAPP (128 μM) and Aβ42 (11 μM) incubations were made as for ThT binding assays (Aβ42 without ThT) and deposited onto nitrocellulose membranes (IAPP: 40 μg, Aβ42: 10 μg) either directly following their preparation (for monomers) or after 2 days of aging (for fibrils; confirmed by ThT binding and TEM). After blocking (5% milk in TBS-T, 2 h, RT) and several washing steps (with TBS-T and ThT buffer), membranes were incubated with N-terminal fluorescein-labeled ACMs (Fluos-ACMs) at 0.2 μM for IAPP-related membranes or 2 μM for Aβ42-related membranes; incubation was overnight (10 °C) in ThT buffer (containing 1% HFIP). To control for fibril autofluorescence, similar membranes were incubated in parallel with buffer only. Bound peptides were visualized using a LAS-400mini instrument (Fujifilm) (software: Image Reader LAS-4000 mini (V2.0)) equipped with a suitable fluorescence filter. Scans of uncropped blots are provided in the Source Data file.

## Dot blot analysis for quantification of fibrils

Dot blot analysis was used to verify the presence of equal amounts of homo- and heteromeric fibrillar assemblies in the aliquots of solutions examined by the various different assays, e.g., the ThT binding assay, the MTT reduction assay, or the PK digestion assay. For example, in the case of the solutions used for ThT binding and MTT reduction assays, 7-day-aged IAPP (16.5 μM) or IAPP/ACM (1/2) mixtures were prepared as described under "ThT binding assays"; TEM showed that fibrils were major species in both kinds of solutions (Fig. 2f). Aliquots (1.3 μg IAPP) were spotted onto 0.2 μm-nitrocellulose membranes. For comparison, freshly made solutions (containing no fIAPP or hf-IAPP/ACM according to ThT binding (IAPP) and TEM) were made as above and spotted immediately. IAPP present in 0 h-aged solutions of IAPP alone and solutions containing non-fibrillar IAPP/ACM co-assemblies (TEM data Fig. 4a) was revealed using a rabbit polyclonal anti-IAPP antibody (Peninsula; 1:1000), whereas for fIAPP present in 7-day-aged solutions the fIAPP-specific mouse anti-fIAPP antibody[23] (Synaptic Systems; Cl. 91E7, 1:500) was used. Incubations with antibodies and membrane development were done under "PK digestion assays". Scans of uncropped blots are provided in the Source Data file.

## Dot blot analysis to detect (TAMRA-)Aβ42 oligomers

The presence of oligomers at various incubation time points of (TAMRA-)Aβ42 (5 μM) solutions (prepared as for the ThT binding assay (Supplementary Fig. 27a) but without ThT or as for the MTT reduction assay (Supplementary Fig. 27b)) was studied by dot blot analysis (Supplementary Fig. 27c). At the indicated time points, Aβ42 (4.5 μg) or TAMRA-Aβ42 (4.9 μg) was spotted onto a nitrocellulose membrane. After blocking (5% BSA in ddH₂O, 2 h, RT) and several washing steps with TBS-T, membranes were incubated overnight (10 °C) with (anti-) oligomer A11 polyclonal antibody (Thermo Fisher Scientific) (1:1000 in 5% BSA in ddH₂O). The primary antibody was combined with donkey anti-rabbit-POD (1:5000); detection as under "Cross-linking, NuPAGE, and Western blot".

## Assessment of fibril thermostability by TEM and ThT binding

Solutions consisting mainly of fIAPP, hf-IAPP/Nle3-VF, fAβ42, or hf-Aβ42/Nle3-VF were prepared as described under "ThT binding assays"

(flAPP 16.5 μM, 7-day-aged; hf-IAPP/Nle3-VF, IAPP 16.5 μM, Nle3-VF 33 μM, 7-day-aged; fAβ42 5 μM, 6-day-aged; hf-Aβ42/Nle3-VF, 5 μM each, 6-day-aged; Aβ42 incubations without ThT) and boiled (95 °C) for 5 min except for fAβ42 which was boiled for 15 min. TEM grids were loaded, stained, and analyzed under "TEM". ThT binding of flAPP and hf-IAPP/Nle3-VF solutions was assessed by mixing aliquots before or after boiling with a ThT solution as described under "ThT binding assays"; buffer values were subtracted from the data.

### Proteinase K (PK) fibril digestion assay in combination with dot blot

The PK digestion assay was performed based on protocols by refs. 71, 72. Briefly, PK stocks (100 μg/ml) were prepared in 50 mM TRIS/HCl pH 8.0 containing 10 mM $CaCl_2$; the final PK concentration in the assay was 0.5 μg/ml. flAPP (16.5 μM) or hf-IAPP/ACM (1/2) were prepared by incubating the peptides in 10 mM sodium phosphate buffer, pH 7.4 for 7 days (20 °C); fibril formation was confirmed by ThT binding (flAPP) and TEM. fAβ42 (5 μM) and hf-Aβ42/ACM (1/1) were made as described under "ThT binding assays" (without ThT; 7-day-aging). For IAPP-related assays, solutions were made by mixing 60 μl of the flAPP or hf-IAPP/ACM solutions with 0.3 μl of the PK stock solution. For Aβ42-related assays, solutions were made by mixing 200 μl of fAβ42 or hf-Aβ42/ACM solutions with 1 μl PK stock. Solutions made as above but without PK were used as controls for 100% undigested fibrils. Solutions were incubated at 37 °C, and at indicated time points, aliquots were dotted onto nitrocellulose membranes, and spots were quickly dried by air. Membranes were washed (TBS-T) and blocked (5% milk in TBS-T, overnight (10 °C)). The following primary antibodies were used for membrane development (2 h, in 5% milk in TBS-T (RT)): mouse anti-flAPP (Synaptic Systems, Cl. 91E7; 1:500) for flAPP[23]; mouse anti-Aβ(1–17) (6E10, BIOZOL; 1:2000) for Aβ42); rabbit anti-Aβ40 (Sigma-Aldrich; 1:2000) for ACMs. Primary antibodies were combined with goat anti-mouse-POD (1:1000) or donkey anti-rabbit-POD (1:5000); detection was as under "Cross-linking, NuPAGE and WB".

### Phagocytosis assay

Phagocytosis of flAPP and fAβ42 versus hf-IAPP/ACM and hf-Aβ42/ACMs was studied in primary murine BMDMs and cultured murine BV2 microglia using TAMRA-IAPP and TAMRA-Aβ42 and essentially following an established protocol[50]. Briefly, BV2 cells (RRID:CVCL_0182) were obtained from Dr. M. Kipp (Rostock University) (who had purchased them from ATCC (EOC2 (CRL-2467)) and were maintained in GlutaMAX-supplemented RPMI1640 medium containing 10% FBS and 1% penicillin/streptomycin on poly-L-ornithine-coated flasks. For the phagocytosis assay, BV2 cells were seeded into 24-well plates containing coverslips in serum-free RPMI1640-GlutaMAX and further incubated for 24 h (5% $CO_2$, 37 °C) to reach 10,000 cells/well. Primary BMDMs were obtained from bone-marrow monocytes isolated from wildtype C57BL/6 mice (8–16 weeks of age, both sexes (Charles River Laboratories)). Ethics oversight by ethical committee on animal care and use of the government of Bavaria (Regierung von Oberbayern, ROB), Germany. Cells were plated in 24-well plates at a density of 10,000 cells/well, and differentiated with L929 cell-conditioned medium (RPMI1640, 10% FBS, 1% penicillin/streptomycin) for 7 days. Thereafter, cells were incubated with aged peptide solutions at 37 °C for 6 h. Peptide solutions were prepared as follows: For IAPP-related studies, TAMRA-flAPP (16.5 μM) and hf-TAMRA-IAPP/ACM (16.5 μM) were prepared by incubating peptides/peptide mixtures (1/2) in ThT buffer for 7 days as for the ThT binding assay. Solutions were diluted with cell medium and added to the cells at a final homo-/hetero-nanofiber (IAPP) concentration of 3.3 μM. For Aβ42-related cell uptake studies, TAMRA-fAβ42 (5 μM) and hf-TAMRA-Aβ42/ACM (5 μM) were prepared by incubating the peptides/peptide mixtures (1/1) under ThT assay conditions (without ThT) for 6 days. Following centrifugation (20 min, 20,000×g), pellets were resuspended in cell medium and

incubated with the cells for 6 h at a final homo-/hetero-fibril concentration of 1 μM. Of note, dot blot analysis and BCA showed that the main peptide fraction was present in the pellet. To address the question of whether the addition of Nle3-VF or L3-VF to preformed Aβ42 oligomers may affect their phagocytosis, preformed TAMRA-Aβ42 oligomers (TAMRA-oAβ42) were prepared as for MTT reduction assays by incubating TAMRA-Aβ42 (5 μM) for 2 h under ThT binding assay conditions. For studying the effect of the ACMs, TAMRA-oAβ42 was mixed with Nle3-VF or L3-VF (1/1) and, following co-incubation for 2 h (ThT binding assay conditions), mixtures were diluted with cell medium (TAMRA-Aβ42, 1 μM) and incubated with the BV2 cells for 6 h as described above. TAMRA-oAβ42 alone was incubated for 2 h as well and treated thereafter as its mixtures with the ACMs.

Following peptide incubation with the cells, supernatants were removed and cells on the coverslips were washed five times with ice-cold 1xPBS, fixed with 4% paraformaldehyde, washed with 1xPBS, permeabilized with 0.2% Triton-X 100, and rinsed three times with cold 1xPBS. Coverslips were mounted with Vectashield Antifade mounting medium containing DAPI (Vector Laboratories). Images were acquired using a Leica DMi8 fluorescence microscope. The percentage of cells that had taken up peptides was calculated by dividing the number of BV2 cells or BMDMs that phagocytosed TAMRA-labeled peptide by the total cell count, multiplied by 100. Significance was analyzed (Graph-Pad 5 and 9) by unpaired student's t-test (2-sided) or one-way ANOVA and Bonferroni as indicated.

### Hippocampal LTP measurements

LTP measurements were performed as previously described in refs. 20, 21, 73, 74. Briefly, sagittal hippocampal slices (350 μm) were obtained from C57BL/6N mice (6–8 weeks of age, male (Charles River Laboratories)) in ice-cold Ringer solution bubbled with a mixture of 95% $O_2$ and 5% $CO_2$ as described in ref. 20 and according to protocols approved by the ethical committee on animal care and use of the government of Bavaria Germany. Ethics oversight by ethical committee on animal care and use of the government of Bavaria (Regierung von Oberbayern, ROB), Germany. Extracellular recordings were performed using artificial cerebrospinal fluid (ACSF)-filled glass microelectrodes (2–3 MΩ) at RT. ACSF consisted of 125 mM NaCl, 2.5 mM KCl, 25 mM $NaHCO_3$, 2 mM $CaCl_2$, 1 mM $MgCl_2$, 25 mM D-glucose, and 1.25 mM $NaH_2PO_4$ (pH 7.3) and was bubbled with 95% $O_2$ and 5% $CO_2$. Field excitatory postsynaptic potentials (fEPSPs) were evoked in the hippocampal CA1 dendritic region via two independent inputs by stimulating the Schaffer collateral commissural pathway (Sccp). For LTP induction, high-frequency stimulation (HFS; 100 Hz/100 pulses) conditioning pulses were delivered to the same Sccp inputs. Both stimulating electrodes were used to utilize the input specificity of LTP, thus allowing for the measurement of internal control within the same slice. Aβ42 (50 nM), Aβ42/ACM mixtures (1/10) or ACMs alone (500 nM) were freshly dissolved in ACSF and applied 60–90 min before HFS. Responses were measured for 60 min after HFS. To study the effects of ACMs on the LTP impairment mediated by pre-oligomerized Aβ42, Aβ42 (50 nM) was pre-incubated in ACSF for 24 h (30 °C) and then mixed with L3-VF or F3-LF (1/10) (in ACSF); pre-oligomerized Aβ42 alone and Aβ42/ACM mixtures were applied to the slices 90 min before HFS[73,74]. fEPSP slope measurements (20–80% of peak amplitude) are presented as % fEPSP slope of baseline (the 20 min control period before tetanic stimulation was set to 100%). Data analysis by one-way ANOVA and Bonferroni's multiple comparison test or by Kruskal–Wallis test with Dunn's multiple comparisons test (GraphPad 6) as indicated.

### X-ray fiber diffraction

flAPP and hf-IAPP/Nle3-VF were made by aging IAPP (1024 μM) or a mixture of IAPP (1024 μM) and Nle3-VF (2048 μM) in ddH₂O for 3 days. A droplet of each solution was placed between glass rods supported by plasticine balls and allowed to dry (humidified atmosphere, 2–4 days,

RT). X-ray diffraction data were collected at the facility Single-Crystal X-Ray Diffractometry of the TUM Catalysis Research Center (CRC) using a Bruker D8 Venture diffractometer equipped with a CPAD detector (Bruker Photon II), an IMS micro source with CuKα radiation (λ = 1.54178 Å) and a Helios optic using the APEX3 software package (Version 2019–1.0, Bruker AXS Inc., Madison, Wisconsin, USA, 2019).

## Reporting summary
Further information on research design is available in the Nature Research Reporting Summary linked to this article.

## Data availability
The data supporting the findings of this study are available within the paper and its Supplementary Information files. Source data are provided with this paper. A.K., J.B., and K.T. are co-inventors of the European patent application 22 158 021.0 (applicant Technical University of Munich) (status: pending) related to ACMs, their hetero-assemblies, and potential biomedical applications. Source data are provided with this paper.

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

## Acknowledgements

This work was supported by Deutsche Forschungsgemeinschaft (DFG) grants SFB1035 B06 to A.K. (in the framework of Collaborative Research Center (CRC) SFB1035, project number 201302640) and SFB1035 Z1,

DFG INST409/209-1 FUGG to J.B., SFB1123-A1/A10 to C.W., SFB1123-Z1 to R.T.A.M., INST409/150-1 FUGG to C.W. and R.T.A.M, and by DFG under Germany's Excellence Strategy within the framework of the Munich Cluster for Systems Neurology EXC 2145 SyNergy-ID 390857198 to J.B. and C.W. C.W. is a van der Laar-Professor of Atherosclerosis. We thank C. Peters, C. Kaiser, and S. Weinkauf for help with TEM. We thank C. Jandl and A. Pöthig and the facility for Single-Crystal X-Ray Diffractometry of the TUM Catalysis Research Center (CRC) for XRD and A. Mitraki for advice on XRD sample preparation. We thank the Bavarian Center for Biomolecular Mass Spectrometry (BayBioMS) at TUM School of Life Science and the Mass Spectrometry facility of the TUM Faculty of Chemistry for MALDI and ESI mass spectrometry. We thank A. Bayer, S. Kalpazidou, and J. Häring for contributions to peptide synthesis and purification, E. Andreetto for the synthesis of rat IAPP and IAPP-GI, and M. Bakou for preliminary studies. We thank J. Buchner and other members of SFB1035 for valuable discussions and support.

## Author contributions

A.K. conceived the project. A.K., K.T., J.B., R.T.A.M., G.R., and M.H. designed experiments with contributions from B.D.V., C.L., O.E.B., Y.T., X.P.-B., M.B., and S.H. K.T., B.D.V., C.L., O.E.B., K.H., Y.T., X.P.B., M.B., S.H., M.O., S.P., L.M., and R.T.A.M. conducted experiments. A.K., K.T., B.D.V., C.L., O.E.B., K.H., Y.T., X.P.B., M.B., S.H., L.M., M.H., G.R., M.O., S.P., C.W., R.T.A.M., and J.B. contributed to data analysis or interpretation. A.K. and K.T. wrote and edited the manuscript with contributions from all authors.

## Funding

## Competing interests

A.K., J.B., and K.T. are co-inventors of the European patent application 22 158 021.0 (applicant Technical University of Munich) (status: pending) related to ACMs, their hetero-assemblies, and potential biomedical applications. The remaining authors declare no competing interests.
