## [Peer Review File · Nature Communications]

REVIEWER COMMENTS

Reviewer #1 (Remarks to the Author):

In my opinion the work presented in the article of Kapurniotu et al. is superb both in terms of peptides design, inhibitory activities results, methodology and biophysical tools used. The objectives, the methodologies and techniques, the analyses of the results and the discussion are very well explained and written.

Kapurniotu et al designed peptides derived from the Abeta40 amyloid core Abeta(15-40) as A β amyloid core mimics (ACMs) and inhibitors of amyloid self-assembly and cross-seeding interactions of IAPP and A β 42.

They based their design on Petkova et al. structures of fibrils of Abeta40 describing a β -strand-loop- β -strand motif with A β (12-22) and A β (30-40) forming the β -strands and Abeta(23-29) for the loop.

13 A β (15-40) analogs containing various different loop tripeptide segments, comprising (Nle)₃, (Leu)₃, (Phe)₃, (Arg)₃, (Gly)₃, or Val-Gly-Ser (control) and one pair of two N-methylated residues were designed, synthesized and studied. They identified six 26-residue peptides as effective amyloid inhibitors of both IAPP and Abeta42. All six ACMs bound IAPP with nanomolar affinity and blocked its cytotoxic amyloid self-assembly with nanomolar IC₅₀ values. In addition, all six ACMs bound Abeta42 with nanomolar affinity and blocked its cytotoxic self-assembly, three of them with nanomolar IC₅₀ values. Ex vivo electrophysiology in murine brains showed a full amelioration of Abeta42-mediated damage of synaptic plasticity by ACMs. Furthermore, ACMs also inhibited reciprocal cross-seeding of IAPP and Abeta42 amyloid self-assembly. ACMs belong to the most effective inhibitors of in vitro amyloid self-assembly of IAPP, Abeta42, or both polypeptides.

Remarkably, ACMs, were non-amyloidogenic in isolation, but they exerted their amyloid inhibitor function via co-assembling with IAPP or Abeta42 into amyloid-like but ThT-invisible and non-toxic nanofibers. The fact that aged mixtures of IAPP with all six ACMs exclusively consisted of fibrillar assemblies indistinguishable from fIAPP fibrils by TEM is not completely unusual. However, the methodologies and techniques used to describe a novel mechanism of amyloid inhibition are impressive and convincing. The identified IAPP/ACM nanofibers had the cross- β amyloid core signature by XRD, but were less ordered than fIAPP according to CD spectroscopy. STED, 2PM, and FLIM-FRET studies suggested that nanofibers and basic parts of their fibrous superstructures consisted of laterally co-assembled, parallel arranged or intertwined/twisted, "protofilament-like" IAPP and ACM stacks. The identified Abeta42/ACM nanofibers had similar widths but 2-4 times greater lengths than fAbeta42. Imaging results were consistent with both axial and lateral co-assembly, the former likely underlying hetero-nanofiber elongation. Remarkably also, IAPP/ACM and Abeta42/ACM nanofibers, on the contrary to pathogenic fIAPP and fAbeta42, showed seeding incompetence, comprised thermolability, proteolytic degradability (demonstrated by Proteinase K fibril digestion assay), and a more efficient phagocytosis

(studied in primary murine BMDMs and cultured murine BV2 microglia using TAMRA-IAPP and TAMRA-A β 42, following an established protocol) than fIAPP and fAbeta42.

In conclusion, in my opinion, the study of cross-seeding of amyloid proteins and the design of peptides inhibiting amyloid aggregation based on these cross-seeding are still in their infancy. This work demonstrate that this strategy to design novel and efficient inhibitors of aggregation is highly promising. The concept of non-toxic fibrillar hetero-assemblies is highly innovative. The article is very well written and pleasant to read. I recommend the publication of this article. I have two remarks :

1- The justification for using Abeta structures of Petkova et al reported in 2002 is not so clear for me. Why not using more recent structures : Schmidt, M. et al. Peptide dimer structure in an A β (1–42) fibril visualized with cryo-EM. Proc. Natl Acad. Sci. USA 112, 11858–11863 (2015); Colvin, M. T. et al. Atomic resolution structure of monomorphic A β 42 amyloid fibrils. J. Am. Chem. Soc. 138, 9663–9674 (2016). Wälti, M. A. et al. Atomic-resolution structure of a disease-relevant A β (1–42) amyloid fibril. Proc. Natl Acad. Sci. USA 113, E4976–E4984 (2016).

Xiao, Y. et al. A β (1–42) fibril structure illuminates self-recognition and replication of amyloid in Alzheimer's disease. Nature Struct. Mol. Biol. 22, 499–505 (2015).

2- As probably the inhibiting properties of ACMs could be explained by preventing the presence of oligomers of hIAPP and abeta42, I suggest to use Mass spectrometry and/or Ion Mobility Spectrometry–Mass Spectrometry to identify the effect of ACMs on the amount of small oligomers as well as on the different conformers of monomers or oligomers, and also to identify complexes of monomers or small oligomers of IAPP and Abeta with ACMS. See for IAPP: J. Am. Chem. Soc. 2014, 136, 660–670; Analyst, 2015, 140, 6990–6999; Front. Cell Dev. Biol. 2021, 9:729001.

Abeta : Analytical Chemistry, Vol. 77, No. 21, November 1, 2005; Anal Bioanal Chem. 2009 Dec;395(8):2509-19; J Am Soc Mass Spectrom. 2018 Apr;29(4):786-795

Reviewer #2 (Remarks to the Author):

The manuscript with the title "Designed peptides as nanomolar cross-amyloid inhibitors acting via supramolecular nanofiber co-assembly" describes the generation of inhibitors for prevention of A β and IAPP self-aggregation and cross-seeding between the two peptides. A β amyloid is present in the brain of patients with Alzheimer's disease, while IAPP forms cytotoxic amyloid in the islets of Langerhans as a consequence of beta-cell stress known to be associated with diabetes. IAPP and A β exhibit size and sequence similarities that facilitate co-aggregation of the peptides. A β residues 15-40 was used as a

template in which chemical modifications were introduced to stabilize the β -sheet structure and suppress endogenous amyloid formation without affecting self/cross assembly propensity. Six inhibitors were identified, which underwent careful characterization, including several biophysical, biochemical, and microscopy techniques. The obtained amyloid inhibitors are unique. Unlike many previous inhibitors that drive aggregation from fibril to non-toxic amorphous aggregates, these inhibitors resulted in nanofibers that lacked several amyloid properties, such as the affinity for Thioflavin T and cytotoxic activity. Furthermore, the nanofibers lacked the typical amyloid seeding capacity.

This is a very well-conducted study with several exciting results supported by multiple analyses. In addition, the obtained data are clearly and correctly reported.

I have only minor points:

Interaction between A β or IAPP and the inhibitors resulted in structures not recognized by Thioflavin T. The amyloid dyes Thioflavin T and Congo red have some tinctorial properties in common. However, Congo red has been suggested to bind to the amyloid fibril in an additional way not shared with Thioflavin T.

Therefore, to exclude the presence of amyloid fibril structures together with the inhibitors, I suggest that aggregates are stained with Congo red.

Also, if the aggregates lack amyloid properties, the wording amyloid-like should be avoided.

Reviewer #3 (Remarks to the Author):

This manuscript reports that constrained peptides designed to mimic the amyloid core of A β (ACMs) are nanomolar inhibitors of IAPP and A β 42 self-assembly and suppress reciprocal cross-seeding. In this manuscript, the authors define the nature of ACMs co-assembled heteromeric nanofibers that exhibit various beneficial features including thermolability, proteolytic degradability and cellular clearance. The experiments are well-thought through and executed, and the data support the conclusions drawn in the manuscript. The conclusions that ACMs are promising leads for anti-amyloid drugs in Alzheimer's disease (AD) and type 2 diabetes are particularly important and will be of interest to the broader community of researchers studying these diseases.

The questions raised below are both conceptual for the approach to investigate the protective effect of ACMs against the main A β aggregated species involved in the pathogenesis of the diseases and technical with regards to the experimental paradigms. Soluble prefibrillar oligomers arising from the aggregation

of the A β 42 have been identified as the the most potent A β neurotoxins implicated in AD. As ex vivo electrophysiology in murine brains showed a full amelioration of the damage of synaptic plasticity induced by A β 42 monomers in the presence of ACMs (fig. 7a), this analysis could be conducted by incubating preformed oligomeric assemblies of A β 42 in the presence of ACMs. Thus, the analysis of cellular clearance of TAMRA-preformed A β 42 oligomers in the presence of ACMs could be investigated in cultured BV2 microglia by fluorescence microscopy according to the quantification of phagocytosis of TAMRA-fA β 42 and hf-TAMRA-A β 42/Nle3-VF(L3-VF) (fig. 7 e).

Reviewer #4 (Remarks to the Author):

To learn more about the molecular architecture of the IAPP/ACM nanofibers the authors employ FLIM-FRET, and use fluorescein as the donor fluorophore (Fluos-Nle3-Vf) and Atto647 as the acceptor fluorophore(TAMRA-IAPP, hf-TAMRA-IAPP, TAMRA-A β 42). I have the following comments and questions regarding these FLIM-FRET experiments.

Major Comments

- In panel j of Fig. 3 where the fluorescence lifetime of Fluos-Nle3-Vf in the absence (donor) versus presence of Atto647-TAMRA-IAPP (donor + acceptor) is presented, it is evident from visual inspection that the unquenched lifetime of Fluos-Nle3-Vf (donor) is approximately 2.2 ns, and as quoted in the results text the quenched lifetime of Fluos-Nle3-Vf (donor + acceptor) is 0.8 ns. This shift in lifetime would correspond to a FRET efficiency = $1 - (0.8 / 2.2) = 0.63$ (i.e., 63 %) instead of 0.85 (i.e., 85 %), as is quoted in the results text, and demonstrated by the FLIM-FRET efficiency histogram presented in the far-right bottom panel of panel j. Why is there this discrepancy? An 85 % FRET efficiency would suggest the unquenched lifetime of Fluos-Nle3-Vf is 5.3 ns, which is too high for fluorescein at a pH of 7 or above. A FRET efficiency of 85 % in any case seems very high, especially for fluorescein and Atto647 which don't have maximal spectral overlap. This comment also applies to the FLIM-FRET efficiency histograms presented in Fig. 4 and Fig. 6.

- The fluorescence lifetime of fluorescein is very sensitive to pH as well as solvent, and it is not entirely clear from the materials and methods text, to what extent this is taken into account when calibrating the donor control (Fluos-Nle3-Vf) for each FRET experiment (i.e., Fluos-Nle3-Vf + Atto647-TAMRA-IAPP in Fig.3, Fluos-Nle3-Vf + Atto647-hf-TAMRA-IAPP in Fig. 4, Fluos-Nle3-Vf + Atto647-TAMRA-A β 42 in Fig. 6, Fluos-Nle3-Vf + Atto647-TAMRA-IAPP in Fig. S13), and to what extent this parameter is monitored across the course of a multi-day experiment (e.g., Fig. 4). It is mentioned that Fluos-Nle3-Vf was calibrated in 1xb (10 mM sodium phosphate buffer pH 7.4). Was this the same solvent used for all FRET

experiments (e.g., FRET experiment with TAMRA-A β 42)? Or was this donor only solvent modified to match each FRET experiment? Does the pH of these sample preparations change across a multi-day incubation / ageing and if so how is this taken into account in Fig.4?

Minor Comments

- The lifetime and FLIM-FRET efficiency maps that correspond to the unquenched decay and lifetime data presented for Fluos-Nle3-Vf in the absence of acceptor (donor) should be presented in either Fig. 3, Fig.4 and Fig. 6 or a supplementary figure

for the majority of the FRET experiments was measured in

- In panel j of Fig. 3 where the fluorescence decay of Fluos-Nle3-Vf in the absence (donor) versus presence of Atto647-TAMRA-IAPP (donor + acceptor) is presented, the fits should be overlaid. This comment also applies to panel d of Fig. 4 and panel f of Fig. 6.

- The lifetime and FLIM-FRET efficiency maps that correspond to the unquenched decay and lifetime data presented for Fluos-Nle3-Vf in the absence of acceptor (donor) should be presented in Fig. 3 (similar to what is done in Fig. 4d).

- In the introduction upon first use the following terms should be defined: confocal laser scanning microscopy (CLSM), stimulated emission depletion (STED) imaging, two photon microscopy (2PM) and fluorescence lifetime imaging microscopy (FLIM) of Förster resonance energy transfer (FRET).

Point-by-Point Response to Reviewers' Comments

Reviewer 1

General comment: *“In my opinion the work presented in the article of Kapurniotu et al. is superb both in terms of peptides design, inhibitory activities results, methodology and biophysical tools used. The objectives, the methodologies and techniques, the analyses of the results and the discussion are very well explained and written. Kapurniotu et al designed peptides derived from the Abeta40 amyloid core Abeta(15-40) as A β amyloid core mimics (ACMs) and inhibitors of amyloid self-assembly and cross-seeding interactions of IAPP and A β 42. They based their design on Petkova et al. structures of fibrils of Abeta40 describing a β -strand-loop- β -strand motif with A β (12-22) and A β (30-40) forming the β -strands and Abeta(23-29) for the loop 13 A β (15-40) analogs containing various different loop tripeptide segments, comprising (Nle)₃, (Leu)₃, (Phe)₃, (Arg)₃, (Gly)₃, or Val-Gly-Ser (control) and one pair of two N-methylated residues were designed, synthesized and studied. They identified six 26-residue peptides as effective amyloid inhibitors of both IAPP and Abeta42. All six ACMs bound IAPP with nanomolar affinity and blocked its cytotoxic amyloid self-assembly with nanomolar IC₅₀ values. In addition, all six ACMs bound Abeta42 with nanomolar affinity and blocked its cytotoxic self-assembly, three of them with nanomolar IC₅₀ values. Ex vivo electrophysiology in murine brains showed a full amelioration of Abeta42-mediated damage of synaptic plasticity by ACMs. Furthermore, ACMs also inhibited reciprocal cross-seeding of IAPP and Abeta42 amyloid self-assembly. ACMs belong to the most effective inhibitors of in vitro amyloid self-assembly of IAPP, Abeta42, or both polypeptides. Remarkably, ACMs, were non-amyloidogenic in isolation, but they exerted their amyloid inhibitor function via co-assembling with IAPP or Abeta42 into amyloid-like but ThT-invisible and non-toxic nanofibers. The fact that aged mixtures of IAPP with all six ACMs exclusively consisted of fibrillar assemblies indistinguishable from fIAPP fibrils by TEM is not completely unusual. However, the methodologies and techniques used to describe a novel mechanism of amyloid inhibition are impressive and convincing. The identified IAPP/ACM nanofibers had the cross- β amyloid core signature by XRD, but were less ordered than fIAPP according to CD spectroscopy. STED, 2PM, and FLIM-FRET studies suggested that nanofibers and basic parts of their fibrous superstructures consisted of laterally co-assembled, parallel arranged or intertwined/twisted, “protofilament-like” IAPP and ACM stacks. The identified Abeta42/ACM nanofibers had similar widths but 2-4 times greater lengths than fAbeta42. Imaging results were consistent with both axial and lateral co-assembly, the former likely underlying hetero-nanofiber elongation. Remarkably also, IAPP/ACM and Abeta42/ACM nanofibers, on the contrary to pathogenic fIAPP and fAbeta42, showed seeding incompetence, comprised thermolability, proteolytic degradability (demonstrated by Proteinase K fibril digestion assay), and a more efficient phagocytosis (studied in primary murine BMDMs and cultured murine BV2 microglia using TAMRA-IAPP and TAMRA-A β 42, following an established protocol) than fIAPP and fAbeta42. In conclusion, in my opinion, the study of cross-seeding of amyloid proteins and the design of peptides inhibiting amyloid aggregation based on these cross-seeding are still in their infancy. This work demonstrate that this strategy to design novel and efficient inhibitors of aggregation is highly promising. The concept of non-toxic fibrillar hetero-assemblies is highly innovative. The article is very well written and pleasant to read. I recommend the publication of this article. I have two remarks:”*

Our Response: We thank the Reviewer for the positive evaluation of our work.

Specific comments:

Comment 1. *“The justification for using Abeta structures of Petkova et al. reported in 2002 is not so clear for me. Why not using more recent structures: Schmidt, M. et al. Peptide dimer structure in an A β (1–42) fibril visualized with cryo-EM. Proc. Natl Acad. Sci. USA 112, 11858–11863 (2015); Colvin, M. T. et al. Atomic resolution structure of monomorphic A β 42 amyloid fibrils. J. Am. Chem. Soc. 138, 9663–9674 (2016). Wälti, M. A. et al. Atomic-resolution structure of a disease-relevant A β (1–42) amyloid fibril.*

Proc. Natl Acad. Sci. USA 113, E4976–E4984 (2016). Xiao, Y. et al. A β (1–42) fibril structure illuminates self-recognition and replication of amyloid in Alzheimer's disease. *Nature Struct. Mol. Biol.* 22, 499–505 (2015).”

Our response: The Reviewer's comment is justified. We used the well-established A β 40 fibril fold of Tycko and coworkers (Petkova et al. PNAS (2002)) as a template because it has been the most commonly applied fold to model A β /IAPP hetero-amyloids (e.g. Berhanu et al. *ACS Chem. Neurosci.* (2013) & Zhang et al. *Phys. Chem. Chem. Phys.* (2015)). Also, since we wished to design A β 40-derived inhibitors, more recently suggested A β 42 structures were not ideally suited as templates. We addressed this issue by adding a sentence and the 2 above references to the Results part (page 5/6) of the revised manuscript.

Comment 2: “As probably the inhibiting properties of ACMs could be explained by preventing the presence of oligomers of hIAPP and abeta42, I suggest to use Mass spectrometry and/or Ion Mobility Spectrometry–Mass Spectrometry to identify the effect of ACMs on the amount of small oligomers as well as on the different conformers of monomers or oligomers, and also to identify complexes of monomers or small oligomers of IAPP and Abeta with ACMS. See for IAPP: *J. Am. Chem. Soc.* 2014, 136, 660–670; *Analyst*, 2015, 140, 6990–6999; *Front. Cell Dev. Biol.* 2021, 9:729001. Abeta: *Analytical Chemistry*, Vol. 77, No. 21, November 1, 2005; *Anal Bioanal Chem.* 2009 Dec;395(8):2509-19; *J Am Soc Mass Spectrom.* 2018 Apr;29(4):786-795.”

Our response: We thank the Reviewer for this valuable suggestion. As mentioned by the Reviewer, our data suggest that inhibitors sequester IAPP and A β 42 from their self-assembly pathway into non-toxic hetero-assemblies and ESI-IMS-MS has previously been successfully applied to identify low MW homo- and hetero-oligomers of IAPP and A β . To address the Reviewer's suggestion, we collaborated with Dr. M. Haslbeck, an MS-expert at the TUM-Department of Chemistry (Chair of Biotechnology/Prof. J. Buchner), and performed first ESI-IMS-MS studies toward the identification of low MW IAPP and ACM homo- or hetero-oligomers suggested to form by our cross-linking and SEC studies (Fig. 2b,c, Suppl. Fig. 7). Our studies identified IAPP homo-di-/tri-/tetramers in IAPP alone, ACM homo-di-/trimers in ACM alone, and IAPP/ACM hetero-di-/tri-/tetramers in IAPP/ACM mixtures, which fitted well the cross-linking and SEC results. In future collaborative work, we are now planning to use ESI-IMS-MS to understand in more detail IAPP/ACM and A β 42/ACM fibrillar co-assembly process. The results of our ESI-IMS-MS studies are included in the Results part (p. 8) and the data are presented in **new Supplementary Figure 8** of the revised manuscript.

Reviewer 2

General comment: “The manuscript with the title “Designed peptides as nanomolar cross-amyloid inhibitors acting via supramolecular nanofiber co-assembly” describes the generation of inhibitors for prevention of A β and IAPP self-aggregation and cross-seeding between the two peptides. A β amyloid is present in the brain of patients with Alzheimer's disease, while IAPP forms cytotoxic amyloid in the islets of Langerhans as a consequence of beta-cell stress known to be associated with diabetes. IAPP and A β exhibit size and sequence similarities that facilitate co-aggregation of the peptides. A β residues 15-40 was used as a template in which chemical modifications were introduced to stabilize the β -sheet structure and suppress endogenous amyloid formation without affecting self/cross assembly propensity. Six inhibitors were identified, which underwent careful characterization, including several biophysical, biochemical, and microscopy techniques. The obtained amyloid inhibitors are unique. Unlike many previous inhibitors that drive aggregation from fibril to non-toxic amorphous aggregates, these inhibitors resulted in nanofibers that lacked several amyloid properties, such as the affinity for Thioflavin T and cytotoxic activity. Furthermore, the nanofibers lacked the typical amyloid seeding capacity. This is a very well-conducted study with several exciting results supported by multiple analyses. In addition, the obtained data are clearly and correctly reported.”

Our Response: We thank the Reviewer for the positive evaluation of our work.

Specific comments: *“I have only minor points.”*

Comment 1: *“Interaction between A β or IAPP and the inhibitors resulted in structures not recognized by Thioflavin T. The amyloid dyes Thioflavin T and Congo red have some tinctorial properties in common. However, Congo red has been suggested to bind to the amyloid fibril in an additional way not shared with Thioflavin T. Therefore, to exclude the presence of amyloid fibril structures together with the inhibitors, I suggest that aggregates are stained with Congo red.”*

Our response: We thank the Reviewer for her/his valuable suggestion. To address this issue, we performed CR binding/staining assays for IAPP/ACM and A β 42/ACM fibrillar co-assemblies. We found that, in contrast to fIAPP and fA β 42, most assemblies present in aged IAPP/ACM and A β 42/ACM mixtures (consisting mostly of fibrils) do not bind CR and are non-birefringent in polarized light upon staining with CR. The results of our studies are included in the Results part (p. 9 & p.13) and the data presented in the two **new Supplementary Figures 10 and 21.**

Comment 2: *“Also, if the aggregates lack amyloid properties, the wording amyloid-like should be avoided.”*

Our response: The Reviewer’s comment is justified. In fact, the specific wording “amyloid-like” is being used to describe “synthetic fibrils with amyloid properties” (P. Westermark et al. *Amyloid* (2005)). According to the most current ISA nomenclature recommendations, “the term ‘amyloid fibril’ should be used for any cross β -sheet fibril” (M.D. Benson et al. *Amyloid* (2020)). Based on the properties of the IAPP/ACM and A β 42/ACM fibrillar co-assemblies and to follow the Reviewer’s suggestion, we thus replaced the wording “amyloid-like” in our manuscript by “amyloid fibril-resembling”. We hope that the Reviewer agrees with this wording.

Reviewer 3

General Comment: *“This manuscript reports that constrained peptides designed to mimic the amyloid core of A β (ACMs) are nanomolar inhibitors of IAPP and A β 42 self-assembly and suppress reciprocal cross-seeding. In this manuscript, the authors define the nature of ACMs co-assembled heteromeric nanofibers that exhibit various beneficial features including thermolability, proteolytic degradability and cellular clearance. The experiments are well-thought through and executed, and the data support the conclusions drawn in the manuscript. The conclusions that ACMs are promising leads for anti-amyloid drugs in Alzheimer’s disease (AD) and type 2 diabete are particularly important and will be of interest to the broader community of researchers studying these diseases.”*

Our response: We thank the Reviewer for the positive evaluation of our work.

Specific comments: *“The questions raised below are both conceptual for the approach to investigate the protective effect of ACMs against the main A β aggregated species involved in the pathogenesis of the diseases and technical with regards to the experimental paradigms.”*

Comment 1: *“Soluble prefibrillar oligomers arising from the aggregation of the A β 42 have been identified as the the most potent A β neurotoxins implicated in AD. As ex vivo electrophysiology in murine brains showed a full amelioration of the damage of synaptic plasticity induced by A β 42 monomers in the presence of ACMs (fig. 7a), this analysis could be conducted by incubating preformed oligomeric assemblies of A β 42 in the presence of ACMs.”*

Our response: We thank the Reviewer for this valuable suggestion. As suggested, we added ACMs (L3-VF and F3-LF) to pre-oligomerized A β 42 and found that they were indeed able to suppress LTP impairment in murine hippocampal slices ex vivo. These results are included in the Results Part (p. 14) and the data are presented in **new Supplementary Figure 26** of the revised manuscript.

Comment 2: “Thus, the analysis of cellular clearance of TAMRA-preformed A β 42 oligomers in the presence of ACMs could be investigated in cultured BV2 microglia by fluorescence microscopy according to the quantification of phagocytosis of TAMRA-fA β 42 and hf-TAMRA-A β 42/Nle3-VF(L3-VF) (fig. 7 e).”

Our response: Thank you again. We performed the proposed experiments and related ones and found that addition of Nle3-VF or L3-VF to preformed cytotoxic A β 42 oligomers (oA β 42) resulted in suppression of fibrillogenesis and cytotoxicity and a pronounced increase of A β 42 phagocytosis. Our results are included in the Results Part (p. 14/15) and the new data are presented in new Supplemental Figure 27. Future studies will aim at understanding the mechanism underlying these findings.

Reviewer 4

General Comment: “To learn more about the molecular architecture of the IAPP/ACM nanofibers the authors employ FLIM-FRET, and use fluorescein as the donor fluorophore (Fluos-Nle3-Vf) and Atto647 as the acceptor fluorophore (TAMRA-IAPP, hf-TAMRA-IAPP, TAMRA-A β 42).”

Our response: We thank the Reviewer for summarizing the fluorophores used in our FLIM-FRET studies, but would have one comment. In his/her summary, the Reviewer mentions that Atto647 was also used as an acceptor fluorophore. However, this is a misunderstanding, as we did not use Atto647 in any of our FLIM-FRET studies. The fluorophore pair used in all FLIM-FRET studies consisted of fluorescein (donor) and TAMRA (acceptor) which have a good spectral overlap. We only used Atto647 for visualizing some of the hetero-assemblies with CLSM, STED, or 2PM (Fig. 3c,d, ex-Supplementary Fig. 9b,c,f (Supplementary Fig. 11b,c,f in revised manuscript), and ex-Supplementary Fig. 11 (Supplementary Fig. 13 in revised manuscript). The information about fluorophores and peptides is shown above the CLSM, STED, or 2PM images in all Figures and Supplementary Figures. In addition, it is included in the Figure legends. Moreover, in the revision process, we rewrote parts of the corresponding experimental part (p. 28-30) and made sure that this information is included as well (p. 29,30). We hope that this information is clearer in the revised version.

Specific comments: “I have the following comments and questions regarding these FLIM-FRET experiments.”

Comment 1: “In panel j of Fig. 3 where the fluorescence lifetime of Fluos-Nle3-Vf in the absence (donor) versus presence of Atto647-TAMRA-IAPP (donor + acceptor) is presented, it is evident from visual inspection that the unquenched lifetime of Fluos-Nle3-Vf (donor) is approximately 2.2 ns, and as quoted in the results text the quenched lifetime of Fluos-Nle3-Vf (donor + acceptor) is 0.8 ns. This shift in lifetime would correspond to a FRET efficiency = $1 - (0.8 / 2.2) = 0.63$ (i.e., 63 %) instead of 0.85 (i.e., 85 %), as is quoted in the results text, and demonstrated by the FLIM-FRET efficiency histogram presented in the far-right bottom panel of panel j. **Why is there this discrepancy?** An 85 % FRET efficiency would suggest the unquenched lifetime of Fluos-Nle3-Vf is 5.3 ns, which is too high for fluorescein at a pH of 7 or above. A FRET efficiency of 85 % in any case seems very high, especially for fluorescein and Atto647 which don't have maximal spectral overlap. **This comment also applies to the FLIM-FRET efficiency histograms presented in Fig. 4 and Fig. 6.**”

Our response: The Reviewer's comment related to the calculated FLIM-FRET efficiency values is justified and we appreciate the detailed and critical review of our FLIM-FRET data.

In fact, the two Fig. parts showing FLIM-FRET efficiency-related data in Fig. 3j (middle panel/right image and right panel/lower part) were incorrect due to a mistake made during the operation of the Leica software. We have now consulted the Leica FLIM-FRET application specialist, repeated the analysis, and found a FLIM-FRET efficiency of ~55% (instead ~85%). In the revised manuscript, we replaced the

two Fig. 3j-parts by the revised Figures and the corresponding FLIM-FRET efficiency values in text and legends by the correct ones.

Also, due to the same software operation issue, the two parts of Fig. 4d showing FLIM-FRET efficiency-related data (middle panel/lower part and right panel/lower part) were incorrect as well. Here re-analysis (as above and, in addition, by using data from a new “experimentally matched” donor (see below under Reviewers’ comment 2)) resulted in a FLIM-FRET efficiency of ~75% (instead ~85%). In the revised manuscript, we replaced the two above mentioned Fig. 4d-parts by the revised Figures and the corresponding FLIM-FRET efficiency values in text and legends by the correct ones.

Regarding the Reviewer’s comment related to the assumed use of a Fluos/Atto647 fluorophore pair in our FLIM-FRET measurements, please see our response to the Reviewer’s general comment above.

A general clarification with regard to the calculation of all FLIM-FRET efficiencies in our manuscript, i.e. the ones shown in Fig. 3j, Fig. 4d, Fig. 6f, and (ex-)Supplementary Fig. 13, which became Supplementary Fig. 17a-c in the revised manuscript: In the Materials & Methods part (p. 28), we mentioned that formula (1) (see below) was used to calculate the FLIM-FRET efficiencies and this formula was applied by the Reviewer as well to control our values. However, our FLIM-FRET efficiencies were not calculated by using formula (1), which is applied for donors exhibiting mono-exponential lifetime decay, but formula (2), which is applied for donors exhibiting multi-exponential lifetime decay (see below), which was the case with our donor (Fluos-Nle3-VF) in absence or presence of acceptor (TAMRA-(f)IAPP or TAMRA-Aβ42).

$$FRET\ Eff(E) = 1 - \frac{\tau_{DA}}{\tau_D} \quad \text{formula (1)}$$

τ : lifetime

τ_{DA} : lifetime of donor + acceptor

τ_D : unquenched donor lifetime

$$FRET\ Eff(E) = 1 - \frac{\tau_{AvAmp}}{\tau_D} \quad \text{formula (2)}$$

τ_{AvAmp} : amplitude weighted average lifetime of quenched donor (undergoing FRET)

τ_D : amplitude weighted average lifetime of the unquenched donor

τ_{AvAmp} was automatically calculated by the Leica FALCON software after fitting according to formula (3):

$$\tau_{AvAmp} = 1 - \frac{\sum A_i \cdot \tau_i}{\sum A_i} \quad \text{formula (3)}$$

τ_{AvAmp} : amplitude weighted average lifetime

A: amplitude

τ : lifetime

In summary, we have used the correct formula (2) to calculate the FLIM-FRET efficiency in all our FLIM-FRET studies, but due to a software application mistake FLIM-FRET efficiency-related results shown in two parts of Fig. 3j and Fig. 4d were incorrect, whereas results shown in Fig. 6f and ex-Supplementary Fig. 13 (Supplementary Fig. 17 in revised version) were correct. In the context of the revision, we

replaced the FLIM-FRET efficiency-related parts of Fig. 3j and 4d and the corresponding FLIM-FRET efficiency values in text and legends by the correct ones.

In addition, FLIM-FRET results shown in Fig. 4d, 6f, and ex-Supplementary Fig. 13 (Supplementary Fig. 17 in revised version) were recalculated using “experimentally matched” donors, as also suggested by the Reviewer (comment 2; see response to Reviewer’s comment 2). The correct data and values are included in the revised manuscript.

Comment 2:

Comment 2a: *“The fluorescence lifetime of fluorescein is very sensitive to pH as well as solvent, and it is not entirely clear from the materials and methods text, to what extent this is taken into account when calibrating the donor control (Fluos-Nle3-Vf) for each FRET experiment (i.e., Fluos-Nle3-Vf + Atto647-TAMRA-IAPP in Fig.3, Fluos-Nle3-Vf + Atto647-hf-TAMRA-IAPP in Fig. 4, Fluos-Nle3-Vf + Atto647-TAMRA-A β 42 in Fig. 6, Fluos-Nle3-Vf + Atto647-TAMRA-flAPP in Fig. S13), and to what extent this parameter is monitored across the course of a multi-day experiment (e.g., Fig. 4).”*

Our response: To address the Reviewer’s comment related to the pH changes of the buffer solutions used for the incubations, we measured the pH of our buffer solutions under the experimental conditions applied for the peptide incubations used for FLIM-FRET analysis and did not observe any time-dependent changes. This is now stated in the Materials & Methods part on p. 29.

Regarding the Reviewer’s comment on effects of pH changes/solvent and how this is taken into account when calibrating the donor control: The effect of solvent, pH, incubation time etc. was taken into account only in the FLIM-FRET analysis shown in Fig. 3j, where the donor-alone incubation conditions (incl. aging time) were the same as the incubation conditions of the donor/acceptor mixture. To address the Reviewer’s comment (see also comment 2b-1), we prepared and analyzed additional donor-alone controls incubated under same experimental conditions as the donor/acceptor mixtures analyzed in Fig. 4d, 6f, and ex-Supplementary Fig. 13, which is Supplementary Fig. 17 in the revised version, and used their lifetime data to recalculate FLIM-FRET efficiencies. Of note, we did not observe significant donor lifetime changes; donor lifetime in all donor-alone solutions was ~2 ns. In the revised version of the manuscript, we now include accordingly revised Fig. 4d, 6f and Supplementary Fig. 17a-c (ex-Supplementary Fig. 13).

Comment 2b: *“It is mentioned that Fluos-Nle3-Vf was calibrated in 1xb (10 mM sodium phosphate buffer pH 7.4).”*

Our response: Yes, this is correct. The donor solution was Fluos-Nle3-VF (33 μ M) in 1xb, 6 day-aged.

Comment 2b-1: *“Was this the same solvent used for all FRET experiments (e.g., FRET experiment with TAMRA-A β 42)? --or was this donor only solvent modified to match each FRET experiment?”*

Our response: The data of the donor solution mentioned in Reviewer’s comment 2b were used for all FLIM-FRET analyses (Fig. 3j, Fig. 4d, Fig. 6f, and (ex-)Supplementary Fig. 13, which became Supplementary Fig. 17a-c in the revised manuscript). To address the Reviewer’s comment, we prepared and analyzed new donor solutions incubated under the same experimental conditions (including the aging time of the solution) as the corresponding donor/acceptor mixtures; see our response to Reviewer’s comment 2a. In the revised version of the manuscript, we now include the accordingly revised Fig. 4d, 6f and Supplementary Fig. 17a-c (ex-Supplementary Fig. 13). The data of the donor alone solutions are presented in new Supplementary Fig. 14, 16, 17d,e, and 23 (see also response to minor comments 1 and 2.)

Comment 2b-2: *“Does the pH of these sample preparations change across a multi-day incubation / ageing and if so how is this taken into account in Fig.4?”*

Our response: No, the pH did not change; please see our response to comment **2a**.

Minor Comments

1) *“The lifetime and FLIM-FRET efficiency maps that correspond to the unquenched decay and lifetime data presented for Fluos-Nle3-Vf in the absence of acceptor (donor) should be presented in either Fig. 3, Fig.4 and Fig. 6 or a supplementary figure for the majority of the FRET experiments was measured in”*

Our response: We followed the Reviewer’s suggestion and added these data for Fig. 3j, 4d, 6f, and Supplementary Fig. 13, which became Supplementary Fig. 17a-c in the revised version, to the Supplementary Information part. The corresponding new Supplementary Figures are: Supplementary Fig. 14, 16, 23, and 17d,e.

2) *“In panel j of Fig. 3 where the fluorescence decay of Fluos-Nle3-Vf in the absence (donor) versus presence of Atto647-TAMRA-IAPP (donor + acceptor) is presented, the fits should be overlaid. This comment also applies to panel d of Fig. 4 and panel f of Fig. 6.”*

Our response: We followed the Reviewer’s suggestion and added these data for Fig. 3j, 4d, 6f, and Supplementary Fig. 13, which became Supplementary Fig. 17a-c in the revised version, to the Supplementary Information part. These data are included in the corresponding new Supplementary Figures (see response to minor comment **2**): Supplementary Fig. 14, 16, 23, and 17d,e.

3) *“The lifetime and FLIM-FRET efficiency maps that correspond to the unquenched decay and lifetime data presented for Fluos-Nle3-Vf in the absence of acceptor (donor) should be presented in Fig. 3 (similar to what is done in Fig. 4d).”*

Our response: These data are presented in Fig. 3j as in Fig. 4d as well.

4) *“In the introduction upon first use the following terms should be defined: confocal laser scanning microscopy (CLSM), stimulated emission depletion (STED) imaging, two photon microscopy (2PM) and fluorescence lifetime imaging microscopy (FLIM) of Förster resonance energy transfer (FRET).”*

Our response: Thank you. As suggested, we defined all above terms upon first use in the introduction part (p. 5).

REVIEWERS' COMMENTS

Reviewer #1 (Remarks to the Author):

The authors further improved their manuscript and took into consideration the suggestions of the reviewers.

In particular, my personal questions or recommendations were fully taken into account. I have no further comments and I am pleased to accept this manuscript for publication.

Reviewer #3 (Remarks to the Author):

The authors have addressed all of my concerns. I find the revised paper to have well justified the conclusions.

Reviewer #4 (Remarks to the Author):

Thanks for the rebuttal and apologies for confusing the acceptor as ATTO647. All my comments regarding FRET efficiency have been addressed. The manuscript looks good.

Point-by-Point Response to Reviewers' Comments

Reviewer 1

"The authors further improved their manuscript and took into consideration the suggestions of the reviewers. In particular, my personal questions or recommendations were fully taken into account. I have no further comments and I am pleased to accept this manuscript for publication."

Our Response: We thank the Reviewer for his/her time and efforts and the positive evaluation of our manuscript.

Reviewer 3

"The authors have addressed all of my concerns. I find the revised paper to have well justified the conclusions."

Our Response: We are glad that the Reviewer's concerns were sufficiently addressed. We thank the Reviewer for the positive evaluation and his/her time and efforts to improve the manuscript.

Reviewer 4

"Thanks for the rebuttal and apologies for confusing the acceptor as ATTO647. All my comments regarding FRET efficiency have been addressed. The manuscript looks good."

Our Response: We are glad that we sufficiently addressed the Reviewer's comments and thank the Reviewer for the positive evaluation and his/her time and efforts to improve the manuscript.